# More Experts Than Galaxies: Conditionally-Overlapping Experts with Biologically-Inspired Fixed Routing

**Sagi Shaier**
Department of Computer Science
University of Colorado Boulder
`sagi.shaier@colorado.edu`

**Francisco Pereira**
Machine Learning Core
National Institute of Mental Health
`francisco.pereira@nih.gov`

**Katharina von der Wense**
Department of Computer Science
University of Colorado Boulder
Institute of Computer Science
Johannes Gutenberg University Mainz
`katharina.kann@colorado.edu`

**Lawrence E Hunter**
Department of Pediatrics
University of Chicago
`lehunter@uchicago.edu`

**Matt Jones**
Department of Psychology and Neuroscience
University of Colorado Boulder
Google DeepMind
`mcj@colorado.edu`

## Abstract

The evolution of biological neural systems has led to both modularity and sparse coding, which enables energy efficiency and robustness across the diversity of tasks in the lifespan. In contrast, standard neural networks rely on dense, non-specialized architectures, where all model parameters are simultaneously updated to learn multiple tasks, leading to interference. Current sparse neural network approaches aim to alleviate this issue but are hindered by limitations such as 1) trainable gating functions that cause representation collapse, 2) disjoint experts that result in redundant computation and slow learning, and 3) reliance on explicit input or task IDs that limit flexibility and scalability. In this paper we propose Conditionally Overlapping Mixture of ExperTs (COMET), a general deep learning method that addresses these challenges by inducing a modular, sparse architecture with an exponential number of overlapping experts. COMET replaces the trainable gating function used in Sparse Mixture of Experts with a fixed, biologically inspired random projection applied to individual input representations. This design causes the degree of expert overlap to depend on input similarity, so that similar inputs tend to share more parameters. This results in faster learning per update step and improved out-of-sample generalization. We demonstrate the effectiveness of COMET on a range of tasks, including image classification, language modeling, and regression, using several popular deep learning architectures. Code can be found here: `https://github.com/Shaier/COMET.git`.

## 1 Introduction

In recent years, there has been a trend towards developing increasingly larger models (OpenAI, 2023a;b; Fedus et al., 2022; Shuster et al., 2022; Chowdhery et al., 2022), driven by the understanding that a neural network's learning capacity depends on its number of parameters (Shazeer et al., 2017). This approach has yielded impressive results in various fields, including computer vision (Dosovitskiy et al., 2021; Kirillov et al., 2023) and language modeling (OpenAI, 2023b; Chowdhery

et al., 2022). However, with such large size come difficulties, including increased training costs and growing requirements for large amounts of memory and storage.

One approach to mitigating some of these challenges is sparsity, where a subset of the model's parameters is selectively utilized in the computational graph. This concept of sparsity has been widely explored in machine learning (Jacobs et al., 1991; Jordan & Jacobs, 1993; LeCun et al., 1989; Zhou et al., 2019; Hoefler et al., 2021; Shazeer et al., 2017; Bair et al., 2024). Researchers have observed significant benefits of sparsity, including reduced inference costs (Han et al., 2016; Shazeer et al., 2017), improved generalization capabilities (LeCun et al., 1989; Jacobs & Burkholz, 2024; Frankle & Carbin, 2019; Paul et al., 2023), enhanced learning efficiency (LeCun et al., 1989), accelerated learning speed (LeCun et al., 1989; Mittal et al., 2022), less interference and forgetting (Raia, 2020), forward knowledge transfer (Andle et al., 2023; Raia, 2020; Yildirim et al., 2023), and compositionality (Pfeiffer et al., 2024).

Early work on sparsity in neural networks focused on simple methods such as Dropout (Srivastava et al., 2014) and L1 regularization (Tibshirani, 1996; Ng, 2004). These and subsequent works explore sparsity at a fine level of granularity, including single parameters (Mallya et al., 2018; Mallya & Lazebnik, 2018), individual neurons (Xu et al., 2024), or CNN filters (Chen et al., 2019). Other research has explored sparsity at the level of whole networks or sub-networks within the mixture of experts (MoE) framework (Shazeer et al., 2017; Bengio, 2013). These methods generally utilize a routing or gating function (Rosenbaum et al., 2017; 2019; Shazeer et al., 2017; Pfeiffer et al., 2024), which decides which parameters or sub-networks of the model to activate based on the input.

However, existing sparse methods have limitations, which we would summarize as five key concerns: Firstly, most approaches rely on trainable gating functions (Shazeer et al., 2017; Mostafa & Wang, 2019; Rasmussen & Ghahramani, 2001; Li et al., 2023; Rahaman et al., 2021; Chen et al., 2019; Shazeer et al., 2018; Zhou et al., 2022; Gururangan et al., 2022; Lin et al., 2021; Ba & Frey, 2013; Bengio et al., 2016; Fedus et al., 2022; Mallya et al., 2018; Mallya & Lazebnik, 2018; Fernando et al., 2017; Keshari et al., 2018). This design choice is problematic for several reasons, including forgetfulness in continual learning (Pfeiffer et al., 2024; Raia, 2020), representation collapse (i.e., degenerate experts; Chen et al., 2023; Pfeiffer et al., 2024), complex training procedures (Rosenbaum et al., 2019), and other issues (Rosenbaum et al., 2017; Pfeiffer et al., 2024; Rosenbaum et al., 2019). Moreover, using non-trainable routing functions can be more effective (Mittal et al., 2022; Muqeeth et al., 2022). Secondly, many state-of-the-art systems employ architectures based on disjoint experts that do not share parameters (Shazeer et al., 2017; Pfeiffer et al., 2024). This design choice can lead to redundancies and may limit generalization; overlap can also be beneficial (French, 1993; Maini et al., 2023). Thirdly, even when experts overlap, it is unclear whether models can effectively learn to map similar inputs to the same experts, potentially resulting in redundancies (Chen et al., 2023) or interference (Pfeiffer et al., 2024). Fourthly, many existing methods require input or task IDs to determine which mask to apply (Mallya et al., 2018; Yang et al., 2020; Masse et al., 2018; Maini et al., 2023; Pes et al., 2024; Mittal et al., 2022; Muqeeth et al., 2022; Kang et al., 2024; Wortsman et al., 2020), which can be restrictive, as meta-information about inputs is rarely available in real-world applications (Aljundi et al., 2019; Ye & Bors, 2022; Wang et al., 2022). Lastly, the number of experts in current systems is limited, often ranging from a few to a couple of thousand (Shazeer et al., 2017; Jiang et al., 2024), which may not be sufficient for complex tasks (Rasmussen & Ghahramani, 2001).

In this paper we introduce Conditionally Overlapping Mixture of ExperTs (COMET), a general deep learning method that induces a modular, sparse architecture in neural networks, with a number of important properties. First, COMET uses a non-trainable gating function, eliminating the need for iterative pruning or continuous sparsification. Instead, we employ a fixed random projection followed by a $k$-winner-take-all cap operation, inspired by the brain's efficient use of a limited number of active cells via lateral inhibition. As in the brain, these mechanisms combine to produce sparse representations with overlap that depends on input similarity (Bruhin & Davies, 2022). Second, COMET does not require fixed specialization of sub-networks or advance knowledge of the active neurons required for each task, enabling more flexibility and adaptability. Third, the number of possible experts in COMET is exponential in the model size, exceeding the limit of a few thousand in recent work, to effectively tackle more complex tasks. Fourth, these experts overlap based on unsupervised information from input similarities. This yields faster learning and improved generalization. It does this without increasing the number of trainable parameters, or requiring input or task IDs to determine which mask to apply.

COMET integrates concepts from diverse research areas into a concise framework: fixed random projection and $k$-winner-take-all from neuroscience, routing functions from modular neural networks, expert-based approaches from the MoE literature, the notion of implicit experts from dynamic neural networks, the integration of sparsity and modularity from conditional computation, input-dependent masking from various deep learning areas, and the importance of active parameter overlap from continual learning. The present paper focuses on learning and out-of-sample generalization in single tasks, but we conjecture COMET's input-dependent sparsity will also yield advantages for settings involving multiple tasks, including transfer learning, continual learning, and robustness to catastrophic forgetting. These will be tested in future work, although we report a preliminary test of transfer learning in appendix A.10.

We validate our approach through experiments on seven diverse tasks, including image classification, language modeling, and regression, demonstrating that our method is applicable to many popular model architectures such as vision transformers, MLP-mixers, GPTs, and standard MLPs, and consistently provides improved performance.

## 2 RELATED WORK

**Sparsity** Sparsity in deep learning refers to systems where not all parameters are active or participating in the computational graph. Fixed sparsity, such as L1 regularization, intends to minimize the number of active parameters through optimizing a continuous loss function. Variable sparsity includes various methods, such as randomness as in Dropout (Srivastava et al., 2014), and input-dependent, such as in sparse MoE (Shazeer et al., 2017). Notably, continuously optimizing deep networks to be sparse is an NP-hard problem (Jacobs & Burkholz, 2024), and pre-defined sparse architectures can be restrictive. To circumvent that, we propose sparsification using a biologically motivated approach of random projection followed by a cap operation, which activates the strongest cells in the network, similar to the sensory system of fruit flies (Bruhin & Davies, 2022).

**Modularity** Modularity is related to sparsity, where information is conditionally routed to a subset of the network's parameters (Pfeiffer et al., 2024; Rosenbaum et al., 2017). Modular methods can be split into two types. Firstly, those with trainable routing functions, including the sparse MoE (Shazeer et al., 2017), adaptively determine the active parameters during training. Although widely used, this approach can lead to representation collapse, forgetfulness, and redundant computation (Pfeiffer et al., 2024). Secondly, fixed routing function methods have been shown to be more effective (Mittal et al., 2022; Muqeeth et al., 2022). However, these methods typically require prior knowledge of module specialization (Pfeiffer et al., 2024; Muqeeth et al., 2022; Mittal et al., 2022), which is often not available in practice. In contrast, our work demonstrates that module specialization can be achieved in a fully unsupervised manner using fixed random projections. Unlike previous studies that employed fixed random projections as routing functions, either with non-overlapping experts (Roller et al., 2021; Chen et al., 2023) or with reduced performance (Bruhin & Davies, 2022), our approach focuses on the more challenging setting of overlapping experts and achieves large performance gains across diverse tasks. Additionally, our expert overlap correlates with input similarity, making it more likely that parameters are shared between similar inputs. This, in turn, enables positive knowledge transfer between items, resulting in faster learning and improved generalization. Moreover, as the experts overlap, our approach benefits from input-dependent sparsity without an explicitly modular architecture (like in MoE).

**Conditional Computation** Conditional computation integrates sparsity and modularity, in that parameters are dropped in a learned and optimized manner, rather than randomly and independently (Bengio, 2013). A notable example is the sparse MoE framework (Shazeer et al., 2017). Our approach diverges from previous work by uniquely integrating several desirable properties, including fixed routing, overlapping expert assignments based on input similarities, mask determination without requiring meta-information, and scalability to an exponentially large number of experts.

**Dynamic Neural Networks** Dynamic neural networks (Han et al., 2021) share similarities with conditional computation. Unlike traditional MoE approaches, dynamic neural networks do not have explicitly defined experts. Instead, they dynamically select units, layers, or components from the main model for each input (Han et al., 2021). Our approach has parallels with some dynamic neural network methods, such as Piggyback (Mallya et al., 2018) and PackNet (Mallya & Lazebnik, 2018), which also do not define explicit experts. However, our method differs in two key ways. Firstly, it

does not require meta-information to determine the expert mask. Secondly, it ensures that any given input always maps to the same expert throughout both training and inference. This contrasts with architectures that use trainable gating functions, where the same input can activate different experts or representations at different stages of learning, which can be seen as a "moving target".

The example-tied dropout scheme of Maini et al. (2023) is another dynamic neural network in which neurons are partitioned into a "generalization" pool that is active for all inputs and a "memorization" pool for which a small subset is active for each input. The assignment of memorization neurons to each input example is random and fixed throughout training. Our approach differs from example-tied dropout in that similar inputs share more active neurons, which facilitates generalization.

The model superposition of Cheung et al. (2019) applies a task-dependent orthogonal transformation to the activation vector at each layer. Assuming the input distribution has effective dimensionality less than that of the activation space (i.e., the layer width), these transformations map different tasks into different subspaces allowing them to be learned semi-independently. Our input-dependent masking also restricts the representation of each example to a subspace, but it does not require task IDs and it leads similar inputs to have more overlapping representations.

## 3 MIXTURE OF EXPERTS AND INPUT-DEPENDENT MASKING

A standard MoE architecture involves a disjoint set of experts and a gate that combines their predictions (Jacobs et al., 1991). For example in the sparse MoE framework proposed by Shazeer et al. (2017), each MoE module consists of $n$ expert networks, $E_1, ..., E_n$, and a gating network, $G$, that outputs a sparse $n$-dimensional vector of mixture weights. The gating and expert networks are all trainable, each with its own set of parameters. The prediction for an input $\boldsymbol{x}$ is $\sum_{i=1}^{n} G(\boldsymbol{x})_i E_i(\boldsymbol{x})$.

Separately, several recent works have proposed versions of *input-dependent masking* (Maini et al., 2023; Mallya et al., 2018; Mallya & Lazebnik, 2018; Yang et al., 2020; Hung et al., 2019). The general framework involves a network with $n$ neurons and a masking function $m \colon \mathcal{X} \to \{0, 1\}^n$ (where $\mathcal{X}$ is the input space). In processing an example $\boldsymbol{x}$, the network's activations are multiplied elementwise with $m(\boldsymbol{x})$. Thus the prediction for $\boldsymbol{x}$ is $F_{m(\boldsymbol{x})}(\boldsymbol{x})$ where $F_{m(\boldsymbol{x})}$ is the function computed by the sub-network corresponding to the mask $m(\boldsymbol{x})$.

Combining these two lines of work, we propose to view the sub-networks defined by input-dependent masking as *overlapping experts*. Any two experts will typically share many active neurons, and hence weights. This is in contrast to standard MoE where the experts learn disjoint sets of parameters. In the overlapping MoE framework, every subset of the full network is a (potential) expert. The sub-network $F_{m(\boldsymbol{x})}$ is an expert for $\boldsymbol{x}$, and it is also a partial expert for any other $\boldsymbol{x}'$ to the degree that $m(\boldsymbol{x})$ and $m(\boldsymbol{x}')$ overlap, as determined by the inner product $m(\boldsymbol{x})^\top m(\boldsymbol{x}')$. In the overlapping MoE framework, the gating network $G$ is replaced by the masking function $m$.

We further propose that similar inputs should map to similar (i.e., more overlapping) experts. This will facilitate generalization because what is learned about one input will selectively generalize to similar inputs. The next section explains how COMET achieves this property using biologically inspired fixed random projections ($\boldsymbol{V}_\ell$ in eq. (4)) and $k$-winner-take-all capping ($C_{k_\ell}$ in eq. (5)).

## 4 CONDITIONALLY OVERLAPPING MIXTURE OF EXPERTS (COMET)

Our proposed COMET method applies to any backbone NN, augmenting it with a second NN called a routing network that computes input-dependent masks for all layers of the backbone network.

We first describe the COMET architecture for the case where the backbone network is an MLP. Let the backbone MLP have $L$ layers, with layer $\ell$ having $N_\ell$ neurons and learnable parameters comprising a weight matrix $\boldsymbol{W}_\ell \in \mathbb{R}^{N_\ell \times N_{\ell-1}}$ and bias $\boldsymbol{b}_\ell \in \mathbb{R}^{N_\ell}$. Then the forward pass of the unmodified MLP is defined by

$$\boldsymbol{a}_\ell = \boldsymbol{W}_\ell \boldsymbol{x}_{\ell-1} + \boldsymbol{b}_\ell \tag{1}$$

$$\boldsymbol{x}_\ell = f(\boldsymbol{a}_\ell) \tag{2}$$

for $1 \leq \ell \leq L$, where $\boldsymbol{a}_\ell$ is the pre-activation at layer $\ell$, $f$ is the elementwise activation function, $\boldsymbol{x}_0$ is the input to the network, and $\boldsymbol{a}_L$ is its output.

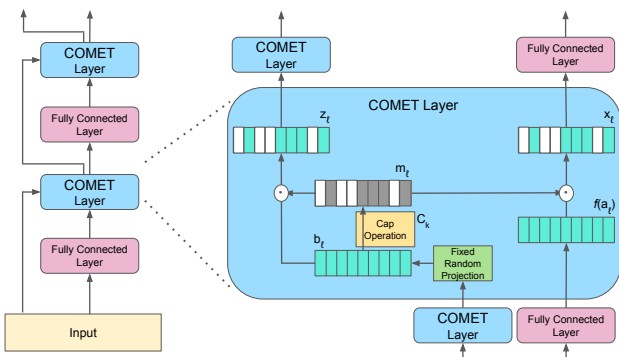

Figure 1: Illustration of a 2-layer MLP with embedded COMET layers. Note that COMET layers do not contain predefined experts, but instead dynamically selects a subset of the backbone MLP's parameters to activate, effectively creating implicit experts. The sparsity level determines the proportion of parameters to activate. Real value in teal, zeros in white, ones in grey.

COMET's routing network is a second MLP with the same shape, defined by random weight matrices $\boldsymbol{V}_\ell$ (for simplicity we omit bias parameters). We sample $\boldsymbol{V}_\ell$ from the same distribution used for initializing $\boldsymbol{W}_\ell$ ($U(-N_{\ell-1}^{-1/2}, N_{\ell-1}^{-1/2})$ in our experiments). We denote this network's pre-activations and activations as $\boldsymbol{c}_\ell$ and $\boldsymbol{z}_\ell$ (analogous to $\boldsymbol{a}_\ell$ and $\boldsymbol{x}_\ell$ in the backbone network), with input $\boldsymbol{z}_0 = \boldsymbol{x}_0$. The computation of the routing network is similar to that of the backbone MLP, except that at each layer it computes a binary vector $\boldsymbol{m}_\ell$ that is then used to mask the activations in both networks. The mask is computed using a $k$-winner-take-all capping function $C_k$:

$$[C_k(\boldsymbol{v})]_i = \begin{cases} 1 & |\{j : v_j \geq v_i\}| \leq k \\ 0 & \text{otherwise} \end{cases} \tag{3}$$

We allow a fixed proportion $p_k$ of neurons at each layer to survive the mask, so that $k_\ell = p_k N_\ell$. Then the forward pass of the routing network is defined by

$$\boldsymbol{c}_\ell = \boldsymbol{V}_\ell \boldsymbol{z}_{\ell-1} \tag{4}$$
$$\boldsymbol{m}_\ell = C_{k_\ell}(\boldsymbol{c}_\ell) \tag{5}$$
$$\boldsymbol{z}_\ell = \boldsymbol{m}_\ell \circ g(\boldsymbol{c}_\ell) \tag{6}$$

where $\circ$ indicates elementwise multiplication and $g$ is the routing network's activation function (we use the identity $g(c) = c$ in the present experiments). The layerwise masks $\boldsymbol{m}_\ell$ computed by the routing network are then applied to the backbone network, so that eqs. (1) and (2) are replaced by

$$\boldsymbol{a}_\ell = \boldsymbol{W}_\ell \boldsymbol{x}_{\ell-1} \tag{7}$$
$$\boldsymbol{x}_\ell = \boldsymbol{m}_\ell \circ f(\boldsymbol{a}_\ell) \tag{8}$$

Note that the network's output $\boldsymbol{a}_L$ is computed before $\boldsymbol{m}_L$ would be applied, avoiding undesirable masking of the model's prediction.

This input-dependent masking results in a maximum number of experts that is exponential in the model size at each layer, specifically $\binom{N_\ell}{k_\ell}$. Therefore in practical settings every input will have its own expert. One consideration for this calculation might be interference among experts. Previous work has studied how multiple models can be superposed within one network (Cheung et al., 2019), and Elhage et al. (2022) show a layer with $N_\ell$ neurons can hold $O(e^N)$ representations that are pairwise orthogonal within a certain finite tolerance, thus minimizing interference. However, more important for the present work is that overlap between models is a desired property because it promotes generalization between similar inputs.

COMET layers differ from sparse MoE (Shazeer et al., 2017) layers in two major ways:

1. **Architecturally**: Whereas a layer in a standard layered MoE architecture consists of $n$ experts and a gating network, a COMET layer contains a random, non-trainable matrix and a $k$-winner-take-all cap operation. Instead of pre-defined experts, COMET layer modifies the computation of the MLP to activate only a subset of its parameters contingent on the input; this subset can be seen as an implicit expert.

2. **In the way the information is passed**: Sparse MoE is applied in layers with a new gating network at each layer, which takes as input the backbone activation at the previous layer. Thus the gating and backbone networks at each layer take the same input. In a COMET architecture the routing network operates independently of the backbone network, so the inputs to the two are distinct (except for the first layer of the network). This ensures that a given example maps to the same implicit expert throughout both training and inference.

Several other important differences between existing approaches and COMET are: 1) COMET's gating network does not require training; 2) COMET does not require fixed specialization of each network module, or advanced knowledge of the combination of modules required for each task; 3) the experts in COMET overlap based on unsupervised information from input similarities; 4) COMET does not require input or task IDs to determine which mask to apply; 5) the number of possible experts in COMET is exponential in the model size.

## 5 EXPERIMENTS

### 5.1 SYNTHETIC DATA EXPERIMENTS

In this section, we describe experiments to verify key properties of a COMET network. First, we verify that the combination of the fixed routing function and cap operator maps similar inputs to similar masks and show how this sharpens the model's generalization. Second, we verify that the network makes an effective use of the available neurons.

### 5.1.1 FIXED INPUT-DEPENDENT ROUTING NETWORK

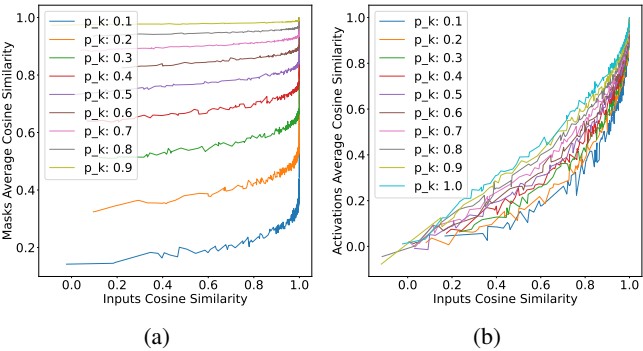

(a)                                         (b)

Figure 2: Routing properties of our gating function, which combines fixed random projections with a cap operator. (a) We compare the similarity of input pairs to the similarity of their corresponding binary masks (gates) for different sparsity levels. This plot shows that similar inputs tend to have similar masks. (b) We compare the similarity of input pairs to the similarity of their corresponding masked activation vectors in the backbone network. This plot reveals that similar inputs are mapped to similar activations, and that this relationship is sharper for sparser networks (note $p_{\mathrm{k}} = 1$ is a vanilla MLP). These properties facilitate forward knowledge transfer, even without supervision.

Our goal is to develop a fixed routing function that maps similar inputs to similar (i.e., overlapping) experts, thereby facilitating knowledge transfer between items and leading to faster learning and improved generalization. One way to approximate generalization between individual training and test items is with the neural tangent kernel (NTK; Jacot et al., 2018). Let $\boldsymbol{\theta} = (\boldsymbol{W}_1, \boldsymbol{b}_1, \ldots, \boldsymbol{W}_L, \boldsymbol{b}_L)$ denote the flattened concatenation of all model parameters, let $\boldsymbol{x}^{\mathrm{train}}$ and $\boldsymbol{x}^{\mathrm{test}}$ be arbitrary training and testing items, and let $\boldsymbol{a}_L^{\mathrm{train}}$ and $\boldsymbol{a}_L^{\mathrm{test}}$ be the corresponding model predictions for some fixed setting of $\boldsymbol{\theta}$. Generalization from $\boldsymbol{x}^{\mathrm{train}}$ to $\boldsymbol{x}^{\mathrm{test}}$ can be defined as the change in prediction $\boldsymbol{a}_L^{\mathrm{test}}$ from including $\boldsymbol{x}^{\mathrm{train}}$ in the training set. Formally, under a vanilla GD optimizer on loss $\mathcal{L}$, and in the limit of a small learning rate $\alpha$, the contribution of $\boldsymbol{x}^{\mathrm{train}}$ to change in $\boldsymbol{a}_L^{\mathrm{test}}$ is

$$\frac{1}{\alpha}\Delta\boldsymbol{a}_L^{\mathrm{test}} \xrightarrow[\alpha\to 0]{} K(\boldsymbol{x}^{\mathrm{train}}, \boldsymbol{x}^{\mathrm{test}})\nabla_{\boldsymbol{a}_L^{\mathrm{train}}}\mathcal{L}^{\mathrm{train}} \tag{9}$$

where $K(\boldsymbol{x}^{\text{train}}, \boldsymbol{x}^{\text{test}})$ is the $N_L \times N_L$ matrix-valued NTK

$$K(\boldsymbol{x}^{\text{train}}, \boldsymbol{x}^{\text{test}}) = \frac{\partial \boldsymbol{a}_L^{\text{test}}}{\partial \boldsymbol{\theta}} \left( \frac{\partial \boldsymbol{a}_L^{\text{train}}}{\partial \boldsymbol{\theta}} \right)^{\top} \tag{10}$$

The RHS of eq. (10) sums over elements of $\boldsymbol{\theta}$, and the contribution from $\boldsymbol{W}_\ell$ is

$$\sum_{ij} \frac{\partial \boldsymbol{a}_L^{\text{test}}}{\partial W_{\ell,ij}} \left( \frac{\partial \boldsymbol{a}_L^{\text{train}}}{\partial W_{\ell,ij}} \right)^{\top} = \sum_j a_{\ell-1,j}^{\text{test}} a_{\ell-1,j}^{\text{train}} \sum_i \frac{\partial \boldsymbol{a}_L^{\text{test}}}{\partial a_{\ell,i}^{\text{test}}} \left( \frac{\partial \boldsymbol{a}_L^{\text{train}}}{\partial a_{\ell,i}^{\text{train}}} \right)^{\top} \tag{11}$$

Thus the contribution of $\boldsymbol{W}_\ell$ to generalization from $\boldsymbol{x}^{\text{train}}$ to $\boldsymbol{x}^{\text{test}}$ is proportional to the inner product $\langle \boldsymbol{a}_{\ell-1}^{\text{train}}, \boldsymbol{a}_{\ell-1}^{\text{test}} \rangle$. This inner product will be positively related to input similarity $\langle \boldsymbol{x}^{\text{train}}, \boldsymbol{x}^{\text{test}} \rangle$ even in an unmodified MLP, but our question is how the relationship changes under COMET.

To answer this question, we conducted an experiment using 500 random pairs of input vectors. Each input had length 100 with components sampled iid from $\mathcal{N}(\mu, 25)$, with $\mu$ a random integer between 0 and 100. We calculated the cosine similarity (i.e., normalized inner product) between each pair of inputs. We then randomly initialized 10 COMET networks for every pair, each comprising a backbone network and a routing network which were both MLPs containing 10 hidden layers ($L = 11$) with 512 neurons per hidden layer. We passed both inputs through each of the 10 COMET networks, using eqs. (4) to (8) with varying degrees of sparsity $p_{\text{k}}$.

To analyze the behavior of our fixed gating function, we performed two complementary analyses. First, we computed the cosine similarity between the masks obtained for the two inputs in each pair, concatenated across layers as $(\boldsymbol{m}_1, \dots, \boldsymbol{m}_{L-1})$. Note that cosine similarity between binary vectors equals their degree of overlap, i.e. the proportion of active neurons for one input that are also active for the other. Second, we measured the cosine similarity between the two inputs' representations in the backbone network after applying the gating function as in eq. (8), again concatenating across layers as $(\boldsymbol{x}_1, \dots, \boldsymbol{x}_{L-1})$. To obtain a more robust estimate, we averaged the cosine similarities across the 10 COMET networks for each input pair, yielding the results in fig. 2.

This experiment reveals that when input distributions are more similar the overlap between their binary masks increases (fig. 2a). This in turn strengthens the relationship between input similarity and activation similarity in the backbone network relative to the baseline MLP with $p_{\text{k}} = 1$ (fig. 2b). Drawing on the NTK analysis above, we conclude that COMET's routing function leads the model to generalize using a narrower effective kernel. A narrower kernel should not be expected to yield universal improvement, but it should be beneficial when the base model has excess capacity for the task. The experiments in the next subsections support this prediction, in that we see an advantage for COMET particularly with larger models.

### 5.1.2 EXPERT UTILIZATION

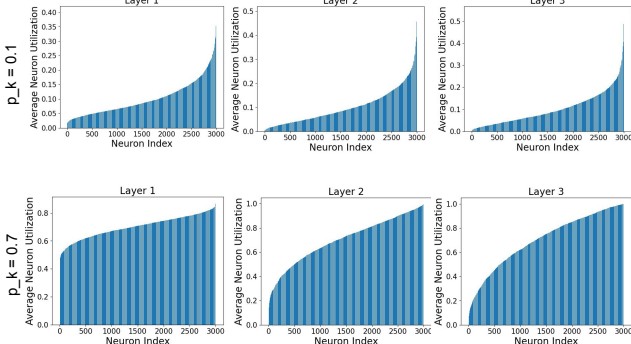

Figure 3: Illustration of neuron activity across COMET layers in a 4-layer MLP. We visualize the utilization of neurons in two randomly initialized networks with varying sparsity levels, using the CIFAR10 dataset. The plots show that our network effectively utilizes all its parameters, with no "dead neurons" and no signs of representation collapse, even at very high sparsity levels.

One of the challenges in training sparse architectures is representation collapse, where a small subset of experts or neurons becomes dominant, leading to under-utilization of others. This issue is particularly concerning when training gating networks, as it can result in "dead" experts or neurons that are never activated. Given that our gating network is fixed, it is essential to investigate whether this behavior occurs, as it would be permanent.

To address this, we conducted an experiment where we generated 1000 randomly initialized 4-layer COMET MLPs with varying neuron counts per layer $N_\ell \sim U(100, 1000)$, and varying sparsity level $1 - p_k \sim U(0.05, 1)$, and passed the CIFAR10 dataset through each one. We then analyzed the utilization of each neuron, measured by how often it is activated with the given masks, and found that only $7.6e^{-4}\%$ of neurons across the ensemble had 0% utilization, and fewer than 2% of the models had any such neurons, mostly models with a high sparsity level. Figure 3 presents utilization plots for two representative networks, illustrating that our fixed gating network avoids representation collapse and "dead neurons." Moreover, when we passed a different dataset (such as CIFAR100) through these models, we discovered that many previously inactive neurons were now being utilized. Thus, neurons that are infrequently utilized appear to be reserved for unseen data, highlighting the network's adaptability and capacity for generalization.

The finding is consistent with our previous analysis in section 5.1.1, which showed that the input-dependent gating design inherently activates similar parameters for similar inputs, facilitating forward knowledge transfer. Our results suggest that our approach effectively mitigates the risk of representation collapse and promotes healthy utilization of neurons in the network, all without relying on supplementary mechanisms, such as specialized loss terms (Shazeer et al., 2017).

## 5.2 IMAGE CLASSIFICATION

We extend our investigation by integrating the COMET method into a diverse range of popular architectures, including Vision Transformers (ViTs), MLP-Mixers, and standard MLPs.

### 5.2.1 STANDARD MLP – CIFAR10

We apply the COMET method to a standard MLP with 4 layers, varying the number of neurons in each layer and the sparsity levels. To evaluate its performance, we compare it to 10 related methods:

**Standard Model:** A standard MLP model with the same number of neurons and no sparsity.

**Smaller Model:** A smaller model with a reduced number of neurons, specifically $p_k N_\ell$ where $N_\ell$ is the width of the standard model.

**Dropout Model:** A standard model with a dropout rate equal to $1 - p_k$.

**Topk Model:** An MLP with a trainable routing function. The cap operation is applied directly to the backbone network by replacing eq. (5) with $\boldsymbol{m}_\ell = C_{k_\ell}(\boldsymbol{a}_\ell)$, so that the routing function selects the highest $k$ values and masks the remaining ones.

**MoE Trainable:** A MoE model with $\lfloor 1/p_k \rfloor$ experts, each having $p_k N_\ell$ neurons in each layer. The routing network is a trainable MLP with one hidden layer and a sparse $\lfloor 1/p_k \rfloor$-dimensional output.

**MoE Non-trainable:** Same as MoE Trainable, with a fixed routing function.

**Layer-wise Routing:** An MLP where each backbone hidden layer representation is projected using a fixed random matrix, which is then used to develop the binary mask for the next layer. This is done by replacing eq. (4) with $\boldsymbol{c}_\ell = \boldsymbol{V}_\ell \boldsymbol{x}_{\ell-1}$.

**Bernoulli Masking:** An MLP where each training example is associated with a fixed binary mask drawn from a Bernoulli distribution, with probability equal to $p_k$. Thus the relationship between inputs and their masks is arbitrary, rather than being mediated by the routing network in COMET.

**Example-tied Dropout:** Example-tied dropout (Maini et al., 2023), where each example in the training data is associated with a fixed binary mask drawn from a Bernoulli distribution, with probability equal to $p_k$, and a fixed number of "generalization neurons" are active for all examples.

**Standard model L1:** A standard MLP model, but using L1 regularization to induce sparsity.

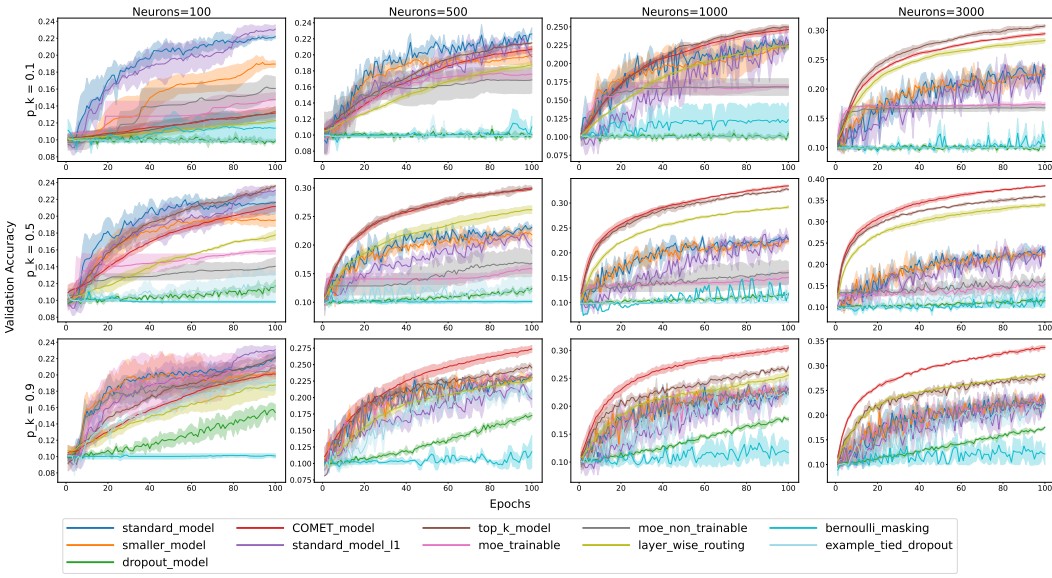

Figure 4: Illustration of 4-layer MLP networks trained on CIFAR10, showcasing the impact of varying network capacity and sparsity levels. As we increase the number of neurons and decrease sparsity (moving from top left to bottom right), we observe a shift in the best-performing model. Initially, the standard model outperforms the COMET model when network capacity is low. However, as network capacity grows, the COMET model emerges as the top performer.

We evaluate these models on the CIFAR10 dataset Krizhevsky (2009), with results shown in fig. 4. Overall, the optimal model architecture depends on the capacity of the network. When the number of neurons is limited and the network has a low capacity to learn the task (i.e., low $p_k$), the standard model that utilizes all neurons outperforms most models. However, as network capacity increases with more neurons, the COMET model emerges as the top performer. This suggests that the benefits of selective neuron activation become more pronounced as capacity increases. Notice in the high-capacity regime the Smaller Model matches the Standard Model, indicating that simply adding more neurons does not improve performance while adding neurons subject to COMET's structured sparsity does.

### 5.2.2 CONTEMPORARY ARCHITECTURES

We further extend the COMET method to contemporary architectures in the Vision domain, including ViT Dosovitskiy et al. (2021) and MLP-Mixer Tolstikhin et al. (2021). To do this, we apply the COMET random projection followed by the cap operation in the MLP layers of each with $p_k = 0.5$. We evaluate the performance of these models on four widely-used image classification datasets: SVHN Netzer et al. (2011), CIFAR10 Krizhevsky (2009), CIFAR100 Krizhevsky (2009), and Tiny ImageNet Le & Yang (2015). Our results can be seen in figs. 5, 6, 10 and 11.

A similar trend emerges in these architectures: as network capacity increases, the optimal model architecture shifts. In smaller networks, where the number of neurons in the MLP layer is limited, the standard model performs roughly similarly to the COMET model. However, even in these networks, incorporating COMET layers yields notable performance improvements. As we scale up the network by adding more neurons, COMET displays superior performance across all five model architectures and four datasets. It achieves faster convergence and significantly higher accuracy, with gains of up to 9% in ViT Large on CIFAR100. Moreover, we observe that the performance gap between the COMET-based models and their standard counterparts widens as the model size increases, with larger models exhibiting both better performance and faster learning rates. This reinforces our finding that selective neuron activation becomes increasingly beneficial as network capacity grows.

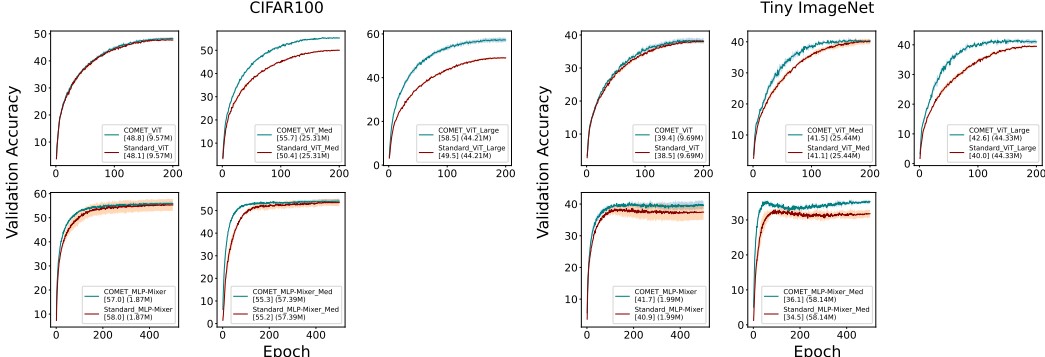

Figure 5: ViTs and MLP-Mixers on CIFAR100. [Highest accuracy] (# trainable param.)

Figure 6: ViTs and MLP-Mixers on Tiny ImageNet. [Highest accuracy] (# trainable param.)

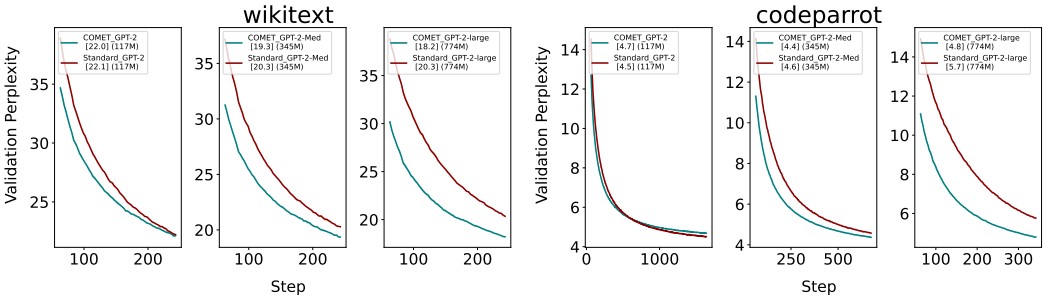

Figure 7: GPTs trained on WikiText. [Lowest perplexity] (# trainable param.)

Figure 8: GPTs trained on CodeParrot. [Lowest perplexity] (# trainable param.)

### 5.3 LANGUAGE MODELING AND REGRESSION

We apply COMET to language modeling on Wikitext (Merity et al., 2016) and CodeParrot (Tunstall et al., 2022) with varying GPT model sizes, with results in figs. 7 and 8. We again observe that as network capacity increases, the COMET model outperforms the standard model, with larger models exhibiting not only a greater performance difference but also faster learning rates, highlighting the benefits of selective neuron activation in language modeling tasks. To further validate our results, we also evaluated COMET on the SARCOS regression dataset. Our findings in appendix A.6 show that our conclusions generalize to this setting as well.

## 6 CONCLUSIONS

In this work, we propose a sparse neural network method, COMET, that induces a modular, sparse architecture with an exponential number of overlapping experts and alleviates key limitations of existing modular approaches, including trainable gating functions that often lead to representation collapse, non-overlapping experts that hinder knowledge transfer, and the need for explicit input IDs. By leveraging a biologically-inspired fixed random projection and $k$-winner-take-all capping operation, COMET determines expert overlap based on input similarity in a fully unsupervised manner, enabling faster learning and improved generalization through enhanced forward transfer. It should be noted that key features of COMET may occur naturally in standard networks. First, similar inputs already tend to have similar hidden representations (see the $p_k = 1$ curve in fig. 2b). Second, there may be cases where networks self-organize to process different inputs in different subspaces of activation space (see Elhage et al., 2022, for a related demonstration). Nevertheless COMET magnifies these effects and our experiments show it improves performance over standard networks. Through extensive experiments on various tasks, including image classification, language modeling, and regression, we demonstrate that COMET achieves improved performance, especially for larger models, demonstrating that our method is applicable to many popular model architectures.

ACKNOWLEDGEMENTS

This research was supported by the Intramural Research Program of the National Institute of Mental Health (NIMH), under project ZIC-MH002968 (PI: Dr. Francisco Pereira), and grant 2020906 from the National Science Foundation (PI: Matt Jones). We would also like to thank Michael Mozer for his invaluable insights and guidance.

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

# A    EXPERIMENT DETAILS

In this section, we provide a detailed description of the experiments conducted to evaluate the performance of COMET. The following subsections outline the experimental setup, including the datasets used, model architectures, and hyperparameters. We also present additional experimental results and analyses.

## A.1    MAIN RESULTS: STANDARD MLP – CIFAR10

In this subsection, we present the main results of our experiments on the CIFAR10 dataset using a standard 4-layer MLP architecture. We first describe the model configuration and training setup, followed by a description of the hyperparameter settings used to vary model capacity and sparsity levels.

**Model Configuration and Training**    We employ a standard 4-layer MLP architecture, utilizing the SGD optimizer with a learning rate of 1e-4. To ensure robustness, we train each model over 3 random seeds for 100 epochs. We systematically explore the effects of varying model capacity and sparsity levels by modifying the number of neurons in each layer and the sparsity ratio.

**Model Capacity Variations**    We consider four different model capacities by setting the number of neurons in each layer to 100, 500, 1000, 3000.

**Sparsity Level Variations**    For each model capacity, we investigate the impact of three different sparsity levels: 0.1, 0.5, and 0.9.

## A.2    ADDITIONAL RESULTS: STANDARD MLP – CIFAR10

This subsection presents additional experimental results on the CIFAR10 dataset using a standard 4-layer MLP architecture, but with a higher learning rate of 1e-3. We describe the model configuration and hyperparameter settings used, and report the results of varying model capacity and sparsity levels.

**Model Configuration and Training**    To further assess the robustness of the COMET method, we also conduct experiments using a higher learning rate of 1e-3. In these experiments, we again employ a standard 4-layer MLP, SGD optimizer, and systematically vary model capacity and sparsity levels by adjusting the number of neurons in each layer and the sparsity ratio, respectively.

**Model Capacity Variations**    We consider an additional model capacity to a total of five different model capacities by setting the number of neurons in each layer to 100, 500, 1000, 3000, or 9000.

**Sparsity Level Variations**    For each model capacity, we investigate the impact of three different sparsity levels: 0.1, 0.5, and 0.9.

As illustrated in fig. 9, a consistent trend emerges as we systematically vary the number of neurons and sparsity levels. Moving from top left to bottom right, we observe a shift in the optimal model configuration. Initially, when network capacity is limited, the standard model outperforms the COMET model. However, as network capacity increases, the COMET becomes the top performer, surpassing every other model. This trend reinforces our key finding: selective neuron activation becomes increasingly beneficial as network capacity increases, enabling faster learning and improved generalization through enhanced forward transfer.

## A.3    CONTEMPORARY ARCHITECTURES

This subsection presents our experimental results on contemporary architectures, including ViT and MLP-Mixer models. We describe the hyperparameter settings used for each architecture, and detail the modifications made to analyze the effect of our sparsity method, COMET.

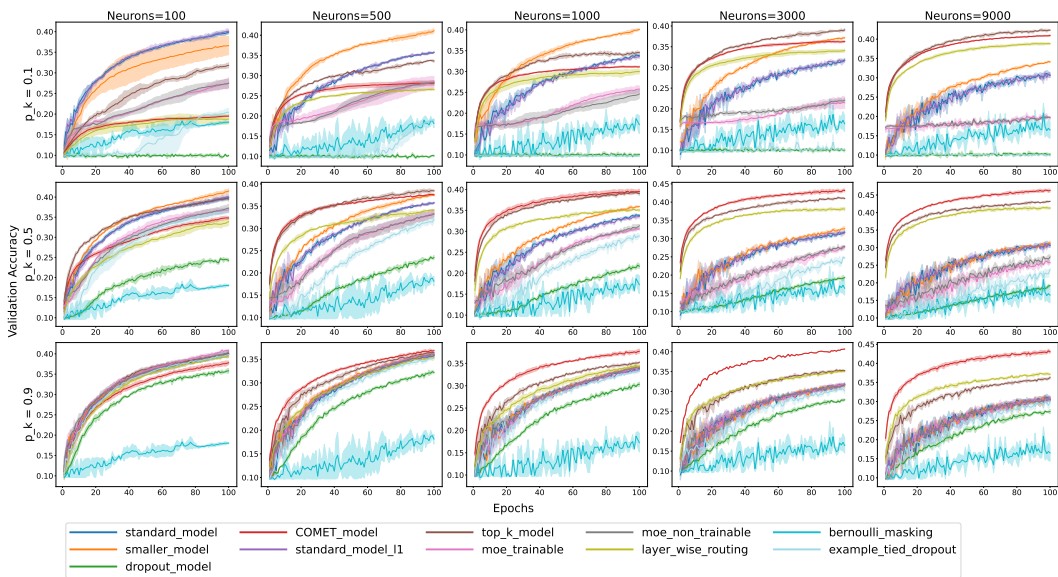

Figure 9: Illustration of 4-layer MLP networks trained on CIFAR10 with a higher learning rate (1e-3), showcasing the impact of varying network capacity and sparsity levels. As we systematically increase the number of neurons and decrease sparsity (moving from top left to bottom right), we observe a shift in the best-performing model. Initially, the standard model (without sparsity, in blue) outperforms the COMET when network capacity is low. However, as network capacity grows, the COMET model emerges as the top performer.

**Hyperparameter Settings**   We built upon the tuned hyperparameters from Yoshioka (2024) and made the following modifications to analyze the effect of our sparsity method, COMET. Specifically, we systematically increased model capacity by adding more neurons to the MLP layers of each model. Additionally, we used the $\tanh$ activation function on the backbone MLP layers. See appendix A.7 for further analysis on activation functions.

**Vision Transformer Hyperparameters**   The standard ViT uses a set of hyperparameters, which we modified as follows. The number of classes is set to 10 for CIFAR10 and SVHN, 100 for CIFAR100, and 200 for Tiny Imagenet. The model depth is 6, with 8 attention heads. We increased the MLP dimension from 512 to 3072 for the ViT medium model and to 6144 for the ViT large model. The dropout rate is 0.1, and the patch size is 4. The embedding dropout rate is also 0.1. We trained the models for 200 epochs with a learning rate of 1e-4.

**MLP-Mixer Hyperparameters**   The MLP-Mixer model uses a different set of hyperparameters. The patch size is 4. We increased the dimension from 512 to 3072 for the MLP-Mixer medium model. The model depth is 6, and the number of classes is set to 10 for CIFAR10 and SVHN, 100 for CIFAR100, and 200 for Tiny Imagenet. We trained the models for 500 epochs with a learning rate of 1e-3.

**Training Setup**   All models were trained using the Adam optimizer with a cosine learning rate schedule on a single A100 GPU. To ensure robustness, we train each model over 3 random seeds.

## A.4   IMAGE CLASSIFICATION

We provide additional experimental results on the image classification task, evaluated on the CIFAR-10 and SVHN datasets, as illustrated in Figure 11 and Figure 10, respectively.

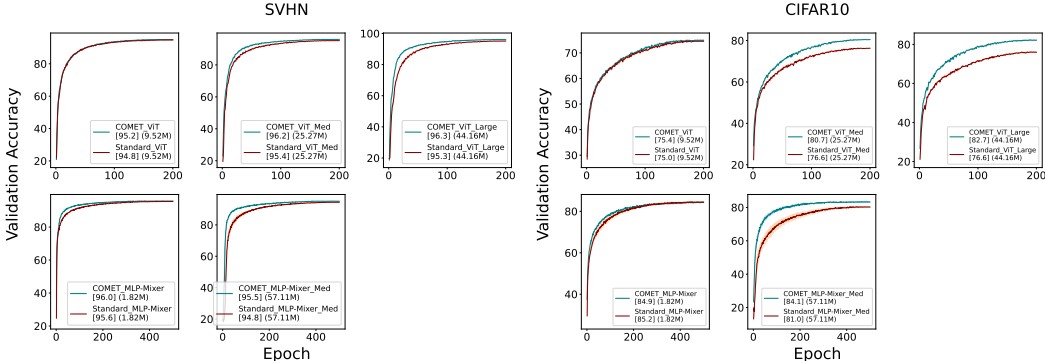

Figure 10: ViTs and MLP-Mixers on SVHN. [Highest accuracy] (# trainable param.)

Figure 11: ViTs and MLP-Mixers on CIFAR10. [Highest accuracy] (# trainable param.)

## A.5 LANGUAGE MODELING

We extend our evaluation of COMET to the task of language modeling, examining its performance on various GPT model variants.

### A.5.1 MAIN RESULTS

This subsection presents our main results on the language modeling task, detailing the performance of COMET on three different GPT model variants. We describe the hyperparameter settings and training setup used for each model, and report the results of our experiments.

**GPT Model Variants** We train three variants of the GPT model, each with a different set of parameters. The standard GPT-2 model has 12 layers, 768 hidden units, 12 attention heads, and 117M parameters. The GPT-2-Medium model has 24 layers, 1024 hidden units, 16 attention heads, and 345M parameters. The GPT-2-Large model has 36 layers, 1280 hidden units, 20 attention heads, and 774M parameters.

**Hyperparameter Settings** We built upon the tuned hyperparameters from HuggingFace (2022). Our optimizer of choice was AdamW, with a learning rate of 5e-4, weight decay of 0.1, and 1,000 warmup steps. We also used gradient accumulation with 8 steps, which resulted in an effective batch size of 256, calculated by multiplying the per-device train batch size (32) by the gradient accumulation steps (8). We used tanh activation function on the backbone MLP layers and a cosine learning rate schedule with warmup. We also enabled mixed precision training to accelerate computations.

**Training Settings** Each model was trained from scratch on a single A100 GPU. Due to computational constraints and time limitations, we restricted training to either 3 epochs or a maximum of 24 hours. To ensure robustness, we train each model over 3 random seeds.

### A.5.2 ADDITIONAL RESULTS

We conducted additional experiments to investigate how the choice of hyperparameters influences the performance of our method, COMET, when applied to the MLP layers of GPT models. This analysis aims to provide a deeper understanding of the robustness and adaptability of COMET under various hyperparameter settings.

The following Figures mark COMET models with spec_true, due to the fact that COMET's input-dependent gating mechanism leads to the formation of experts that are selectively specialized for specific inputs. The standard models are marked with spec_false.

**Tokenizer Effect** We note that, in general, the GPT models we evaluated tend to learn faster when using the tokenizer provided by HuggingFace (2022). However, due to its widespread adoption, we opt to use the standard GPT-2 tokenizer for the remainder of our experiments:

```
tokenizer = AutoTokenizer.from_pretrained("gpt2")
```

To validate our main findings, we first assess the performance of the different GPT-2 model sizes on WikiText and CodeParrot using the standard GPT-2 tokenizer and $50\%$ sparsity level. Our results are presented in figs. 12 to 17.

Consistent with our main results, we find that the COMET-based model learns faster, even with a smaller model size, when using the standard GPT-2 tokenizer. This suggests that the observed pattern is robust and not specific to the tokenizer used.

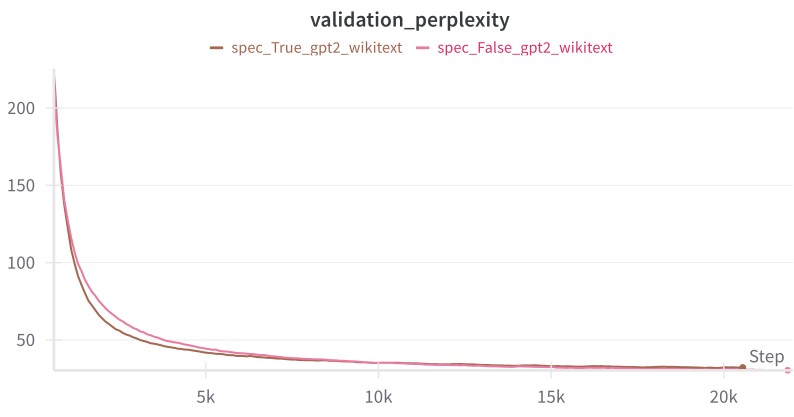

Figure 12: Validation perplexity of GPT-2 on Wikitext dataset using the standard GPT-2 tokenizer.

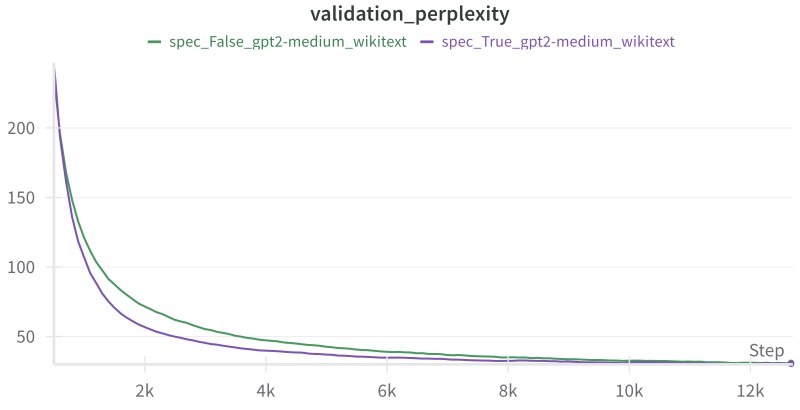

Figure 13: Validation perplexity of GPT-2 Medium on Wikitext dataset using the standard GPT-2 tokenizer.

**Learning Rate Effect**  To investigate the impact of learning rate on the COMET method, we scale the learning rate by a factor of 10, from 1e-4 to 1e-3. The results of this experiment are presented in figs. 18 to 23.

Our findings show that, across all experiments, the COMET models consistently learn faster than their standard counterparts. However, at the $50\%$ sparsity level, we observe that the smaller models often perform slightly worse than the standard models. This result is consistent with our previous findings, which suggest that adding sparsity to models with limited capacity can negatively impact performance.

Furthermore, we identify an interesting trend as the model size increases, as seen in figs. 19, 20, 22 and 23. Specifically, when using a larger learning rate and larger model sizes, the standard models

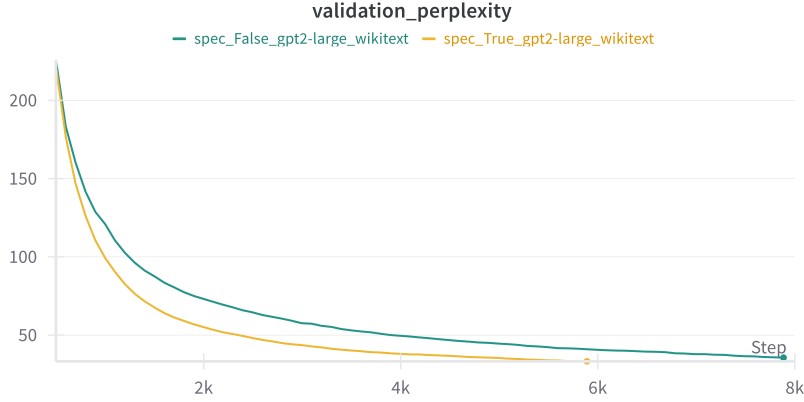

Figure 14: Validation perplexity of GPT-2 Large on Wikitext dataset using the standard GPT-2 tokenizer.

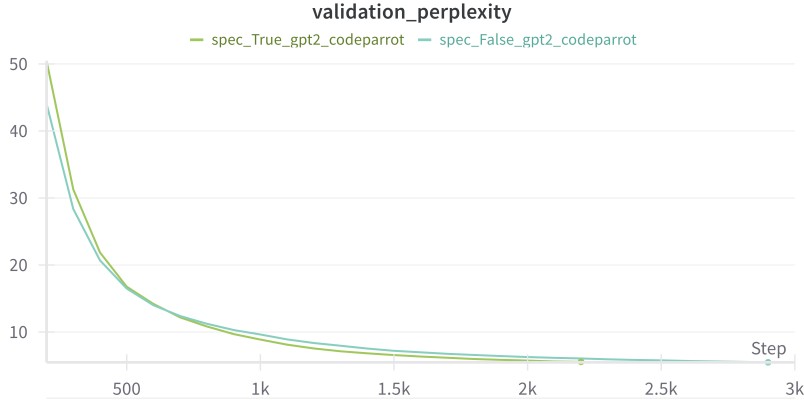

Figure 15: Validation perplexity of GPT-2 on CodeParrot dataset using the standard GPT-2 tokenizer.

tend to overfit or experience exploding gradients, resulting in a significant increase in validation perplexity. In contrast, adding sparsity using COMET not only enables faster learning but also mitigates overfitting and gradient explosion, allowing for stable training with a larger learning rate.

**Batch Size Effect**    To further investigate the robustness of the COMET method, we examine the effect of reducing the batch size by a factor of 4, achieved by decreasing the gradient accumulation step from 8 to 2. Our results are presented in figs. 24 to 29.

Our findings demonstrate that COMET remains effective even with a reduced batch size, making it a viable option for users with limited computational resources. We observe that on the Wikitext dataset, the smaller GPT model with COMET learns faster and achieves comparable performance to the standard model at the end of training. In contrast, on the CodeParrot dataset, the smaller model with COMET outperforms the standard model. Moreover, we note that COMET's benefits extend to smaller batch sizes, where standard models may struggle with overfitting or exploding gradients. By incorporating sparsity, COMET enables more stable training and better performance, even in resource-constrained environments.

**Mixed Precision Effect**    We further investigate the robustness of the COMET method by evaluating its performance under mixed precision training. Specifically, we assess the impact of switching from FP16 to FP32 precision on the COMET-based models. Our results are presented in figs. 30 to 32.

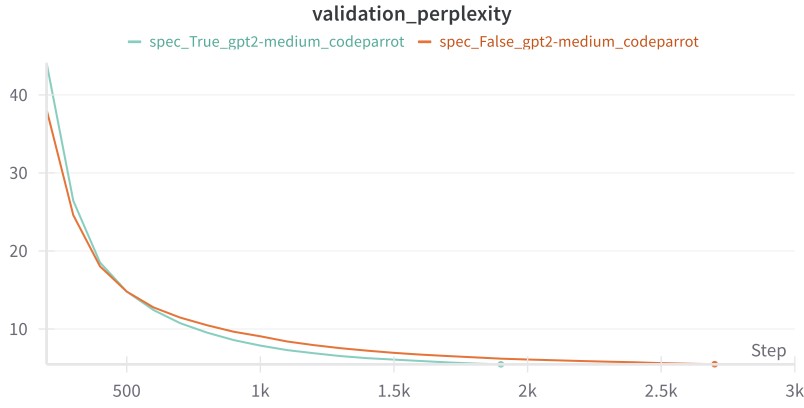

Figure 16: Validation perplexity of GPT-2 Medium on CodeParrot dataset using the standard GPT-2 tokenizer.

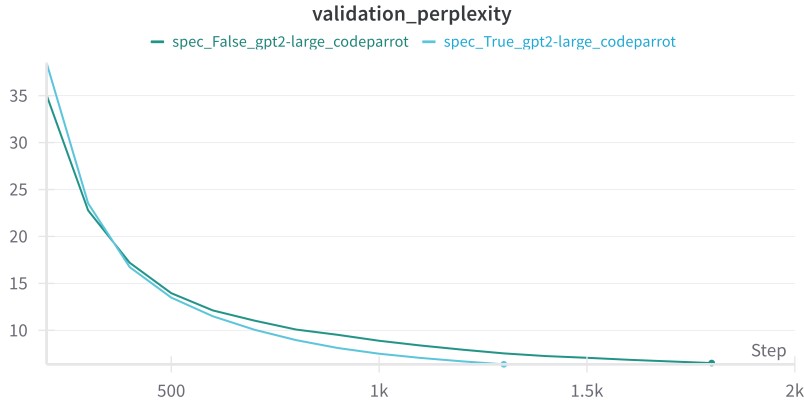

Figure 17: Validation perplexity of GPT-2 Large on CodeParrot dataset using the standard GPT-2 tokenizer.

In summary, our experiments demonstrate that the COMET method consistently enables faster learning across various model sizes, datasets, and training settings. By incorporating COMET, we observe improved performance and robustness, even when switching from FP16 to FP32 precision. This is particularly valuable, as FP32 precision is often preferred in certain applications where numerical stability is crucial. Moreover, having models that can effectively learn in both FP16 and FP32 precision regimes provides greater flexibility and adaptability, allowing for more efficient deployment on a wide range of hardware platforms.

### A.6 REGRESSION

This subsection presents our evaluation of COMET on a regression task using the SARCOS dataset. We describe the experimental setup, including the dataset, model architecture, and hyperparameter settings, and report the results of our experiments.

**Dataset** To conclude our evaluation, we apply the COMET method to a regression task using the SARCOS dataset. This dataset is derived from an inverse dynamics problem involving a 7-joint anthropomorphic robot arm, where the goal is to predict the 7 joint torques based on a 21-dimensional input space consisting of joint positions, velocities, and accelerations. We focus on a single output dimension, following the approach of Rasmussen & Williams (2006). The dataset

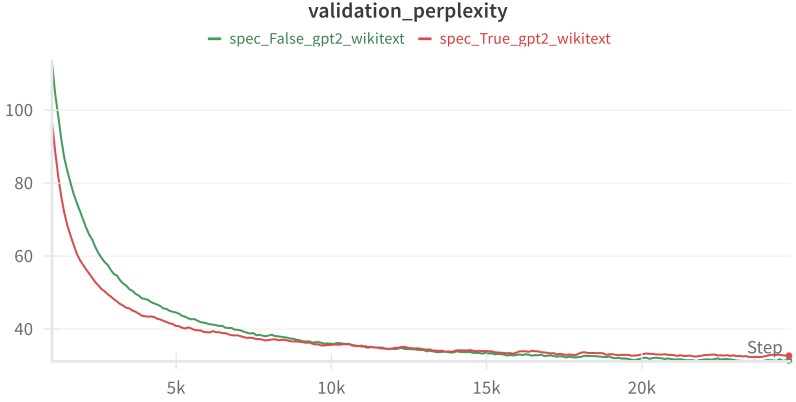

Figure 18: Validation perplexity of GPT-2 on Wikitext dataset using the standard GPT-2 tokenizer and a learning rate of 1e-3.

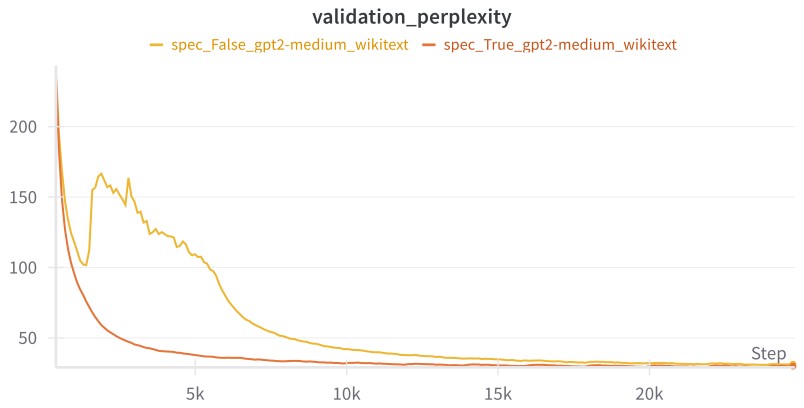

Figure 19: Validation perplexity of GPT-2 Medium on Wikitext dataset using the standard GPT-2 tokenizer and a learning rate of 1e-3.

is publicly available at `https://gaussianprocess.org/gpml/data/`. Building on the work of Jones et al. (2024), we employ a 4-layer MLP network, but experiment with varying the number of neurons in each layer and the level of sparsity.

We use SGD optimizer with a learning rate of 1e-2. To ensure robustness, we train each model over 3 random seeds for 50 epochs. We systematically explore the effects of varying model capacity and sparsity levels by modifying the number of neurons in each layer and the sparsity ratio.

**Model Capacity Variations** We consider five different model capacities by setting the number of neurons in each layer to 100, 500, 1000.

**Sparsity Level Variations** For each model capacity, we investigate the impact of three different sparsity levels: 0.1, 0.5, and 0.9.

**Results** The results, presented in fig. 33, demonstrate the effectiveness of our approach in this domain. Consistent with our previous findings, we again observe that the COMET model outperforms the standard model as network capacity increases, confirming the benefits of selective neuron activation in regression tasks.

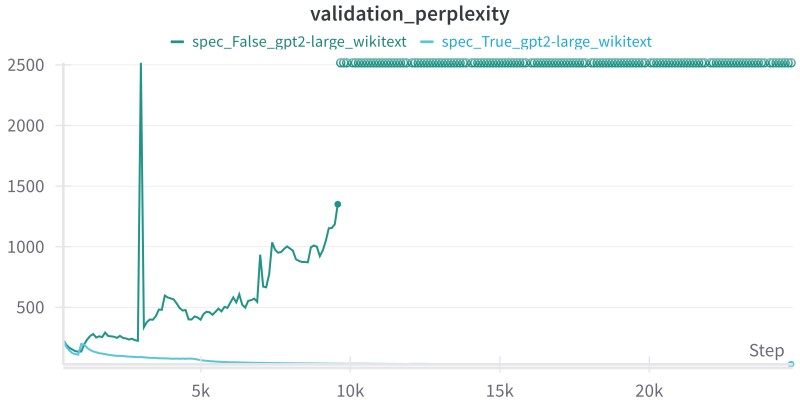

Figure 20: Validation perplexity of GPT-2 Large on Wikitext dataset using the standard GPT-2 tokenizer and a learning rate of 1e-3.

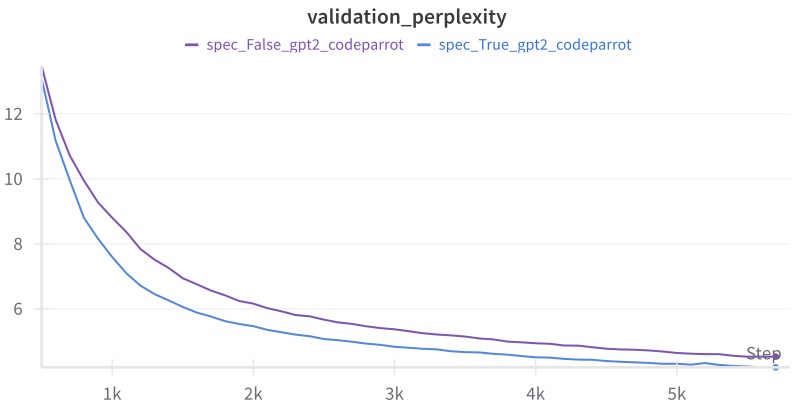

Figure 21: Validation perplexity of GPT-2 on CodeParrot dataset using the standard GPT-2 tokenizer and a learning rate of 1e-3.

### A.7 ACTIVATIONS ANALYSIS

In this section, we analyze the impact of different activation functions on the performance of COMET-based models. We present an experimental evaluation of various activation functions on a 4-layer MLP network trained on the CIFAR10 dataset, and discuss the results.

**Evaluating the Impact of Activation Functions on COMET** We now investigate the impact of utilizing various activation functions within the backbone network on the performance of a COMET-based model during training.

**Experimental Setup** We evaluate the effect of different activation functions on a 4-layer MLP network trained on the CIFAR10 dataset with the following settings: SGD + Momentum optimizer, a learning rate of 1e-3, a sparsity level of 0.5, and 3000 neurons in each layer.

**Results** Our results are presented in fig. 34. We find that the COMET model outperforms the standard model when most activation functions are applied on the backbone network. However, we observe that COMET does not perform as well as the standard model for certain non-monotonic activation functions, specifically GELU, Mish, and SiLU, under the settings we chose (sparsity level = 0.5 and 3000 neurons in each layer). Due to time constraints, we do not delve deeper into the

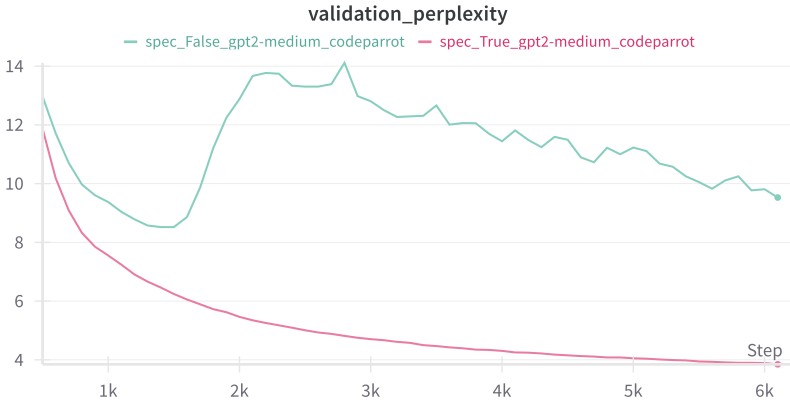

Figure 22: Validation perplexity of GPT-2 Medium on CodeParrot dataset using the standard GPT-2 tokenizer and a learning rate of 1e-3.

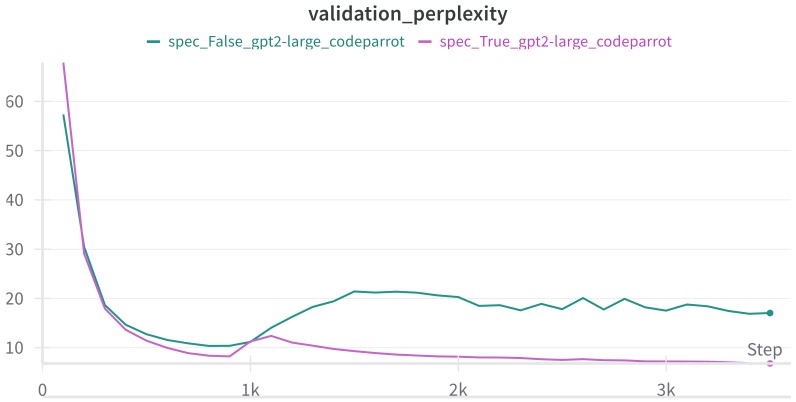

Figure 23: Validation perplexity of GPT-2 Large on CodeParrot dataset using the standard GPT-2 tokenizer and a learning rate of 1e-3.

reasons behind this phenomenon but suggest that future work could investigate why non-monotonic functions may hinder the effectiveness of COMET.

## A.8 COMET: ROUTING NETWORK ARCHITECTURE

We have implemented COMET using a routing network having the same architecture as the backbone MLP network, but this is not necessary. COMET only requires the routing network to generate a mask at each layer having the same shape as (or a shape that can be broadcast to) the shape of the corresponding backbone layer. Future work will investigate alternative routing network architectures to determine if they can improve the performance and efficiency of COMET-based models.

## A.9 RUNNING TIME AND MEMORY REQUIREMENTS

While COMET does not introduce additional trainable parameters, it does require some extra computation to calculate the masks. We evaluate the training times and GPU memory usage of COMET compared to standard vision models.

Tables 1 and 2 show that while COMET introduces some additional computation, the overhead is moderate. For example, the added GPU usage ranges from 0.3 GB (4%) to 7 GB (54%), where the

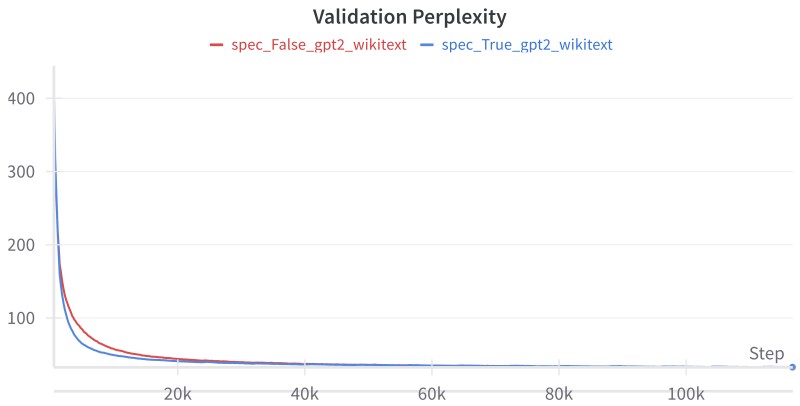

Figure 24: Validation perplexity of GPT-2 on Wikitext dataset using the standard GPT-2 tokenizer and a reduced batch size of 64 ($\frac{1}{4}$ of original).

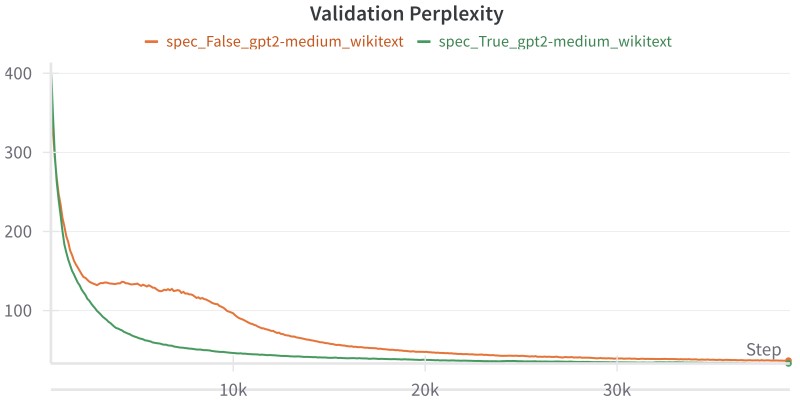

Figure 25: Validation perplexity of GPT-2 Medium on Wikitext dataset using the standard GPT-2 tokenizer and a reduced batch size of 64 ($\frac{1}{4}$ of original).

larger percentage differences occur in cases where the ViT network is excessively overparameterized (e.g., when the number of neurons in the original network is multiplied by 12, from 512 to 6144).

It is worth noting that for simplification, our implementation has not been optimized:

1. We did not use sparse matrix multiplication methods, which could avoid unnecessary computations through the zero masks. This would reduce both the training time (by preventing gradient calculation of the zero gates) and overall GPU usage and inference time (due to less computation). This optimization could lead to significant improvements, potentially even compared to the base models.

2. As mentioned in appendix A.8, we focus on a specific case where the fixed matrix $C_i$ has the same dimensions as the weight matrix $W_i$ at each layer. This means that each random projection matrix has the same shape as the corresponding FC layer's matrix. Notably, however, the only requirement is that they share the same input and output dimensions, leaving room for future exploration of alternative configurations.

Overall, these results suggest that COMET's additional computational overhead is manageable, even with our simple implementation, and that its benefits can be achieved without significantly sacrificing efficiency.

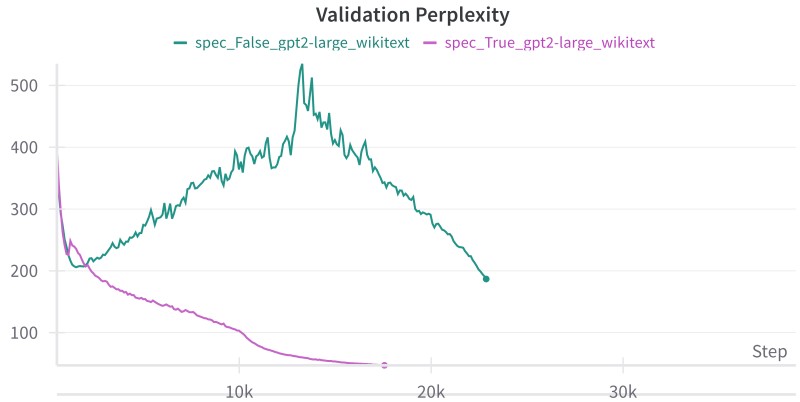

Figure 26: Validation perplexity of GPT-2 Large on Wikitext dataset using the standard GPT-2 tokenizer and a reduced batch size of 64 ($\frac{1}{4}$ of original).

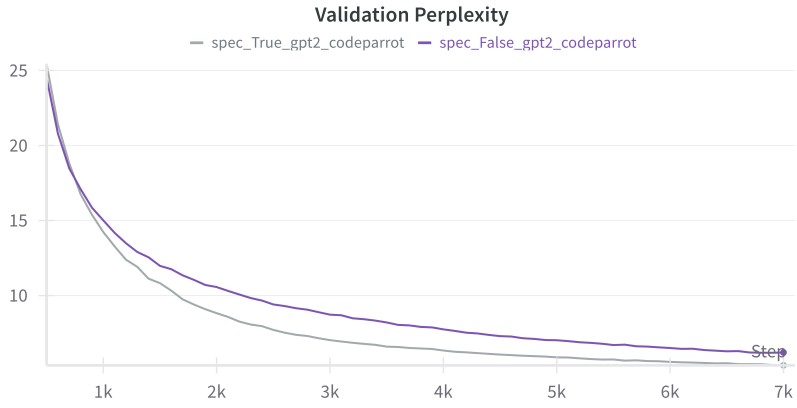

Figure 27: Validation perplexity of GPT-2 on CodeParrot dataset using the standard GPT-2 tokenizer and a reduced batch size of 64 ($\frac{1}{4}$ of original).

### A.10 TRANSFER LEARNING

To further demonstrate COMET's ability to transfer learned knowledge, we conducted an additional experiment. We trained MLP COMET and standard MLP models on CIFAR-10 and then fine-tuned them on SVHN. The results, presented in fig. 35, show that COMET exhibits superior transfer learning capabilities. Specifically, "COMET with transfer" outperforms "standard model with transfer"

| Dataset / Model | SVHN | CIFAR10 | CIFAR100 | Tiny ImageNet |
|---|---|---|---|---|
| COMET ViT | 6.6 | 6.6 | 6.6 | 6.6 |
| Standard ViT | 6.1 | 6.1 | 6.1 | 6.1 |
| COMET ViT-Med | 12.3 | 12.3 | 12.3 | 12.3 |
| Standard ViT-Med | 9.2 | 9.2 | 9.2 | 9.2 |
| COMET ViT-Large | 19.8 | 19.8 | 19.8 | 19.8 |
| Standard ViT-Large | 12.8 | 12.8 | 12.8 | 12.8 |
| COMET MLP-Mixer | 6.7 | 6.7 | 6.7 | 6.7 |
| Standard MLP-Mixer | 6.4 | 6.4 | 6.4 | 6.4 |
| COMET MLP-Mixer-Med | 25.5 | 25.5 | 25.5 | 25.5 |
| Standard MLP-Mixer-Med | 23.2 | 23.2 | 23.2 | 23.2 |

Table 1: GPU Utilization in GBs. All models were trained on a single A100 GPU.

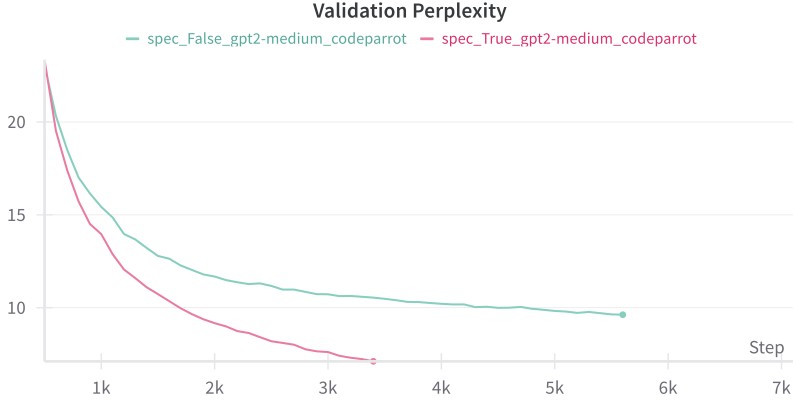

Figure 28: Validation perplexity of GPT-2 Medium on CodeParrot dataset using the standard GPT-2 tokenizer and a reduced batch size of 64 ($\frac{1}{4}$ of original).

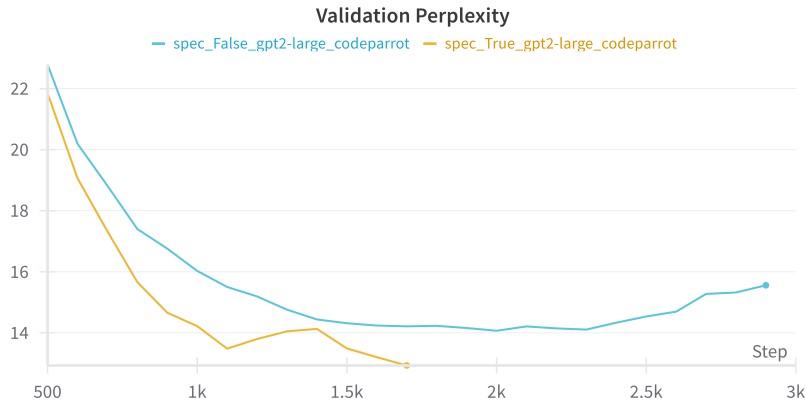

Figure 29: Validation perplexity of GPT-2 Large on CodeParrot dataset using the standard GPT-2 tokenizer and a reduced batch size of 64 ($\frac{1}{4}$ of original).

in both initial accuracy and learning speed. Moreover, "COMET with transfer" also surpasses the overall performance of both "COMET without transfer" and "standard model without transfer" on SVHN. This suggests that COMET's advantage is not limited to out-of-sample generalization in the same distribution but also extends to transfer across tasks.

| Dataset / Model | SVHN | CIFAR10 | CIFAR100 | Tiny ImageNet |
|---|---|---|---|---|
| COMET ViT | 1.1 | 0.8 | 0.8 | 1.2 |
| Standard ViT | 0.9 | 0.7 | 0.7 | 1.2 |
| COMET ViT-Med | 2.0 | 1.5 | 1.5 | 2.4 |
| Standard ViT-Med | 1.3 | 1.1 | 1.1 | 1.3 |
| COMET ViT-Large | 3.3 | 2.2 | 2.0 | 3.3 |
| Standard ViT-Large | 1.8 | 1.4 | 1.3 | 1.7 |
| COMET MLP-Mixer | 1.8 | 1.5 | 1.4 | 2.3 |
| Standard MLP-Mixer | 1.6 | 1.4 | 1.3 | 2.3 |
| COMET MLP-Mixer-Med | 10.2 | 5.4 | 6 | 9.3 |
| Standard MLP-Mixer-Med | 7.6 | 4.5 | 4.8 | 8.1 |

Table 2: Training times in hours. All models were trained on a single A100 GPU.

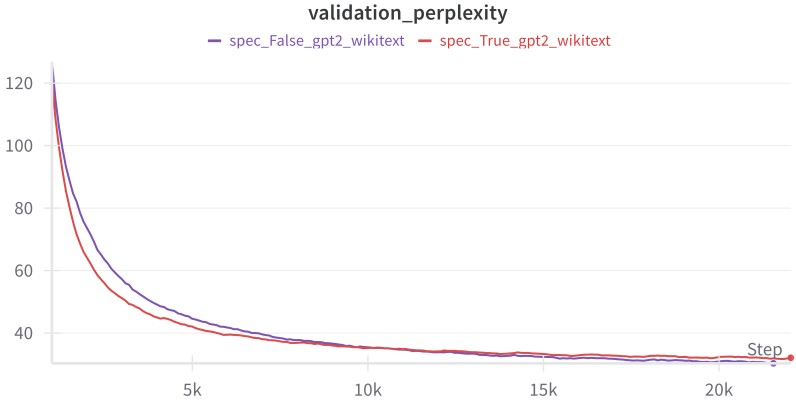

Figure 30: Validation perplexity of GPT-2 on Wikitext dataset using the standard GPT-2 tokenizer and FP32 precision.

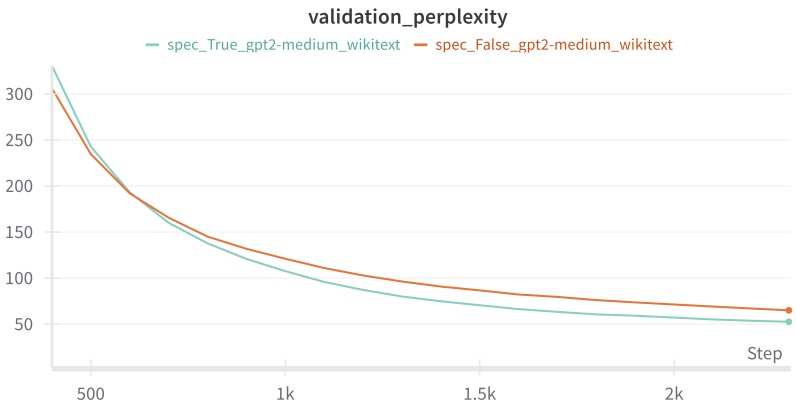

Figure 31: Validation perplexity of GPT-2 Medium on Wikitext dataset using the standard GPT-2 tokenizer and FP32 precision.

Interestingly, we observe a larger gap in performance between "standard model with transfer" and "standard model without transfer" compared to the gap between "COMET model with transfer" and "COMET model without transfer". This might suggest that the standard model benefits more from transfer learning on non-iid data. However, since overall performance is generally taken as the primary metric for evaluating transfer learning, COMET's superior performance on SVHN is the most relevant outcome. This finding is promising for future work that will more thoroughly investigate COMET in multi-task settings including transfer learning and continual learning.

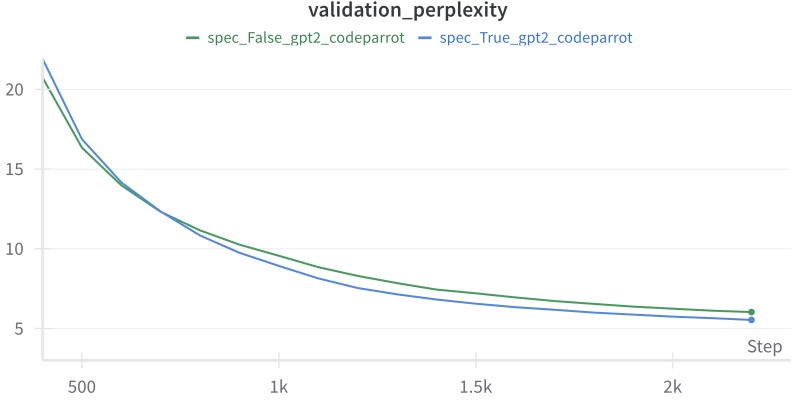

Figure 32: Validation perplexity of GPT-2 on CodeParrot dataset using the standard GPT-2 tokenizer and FP32 precision.

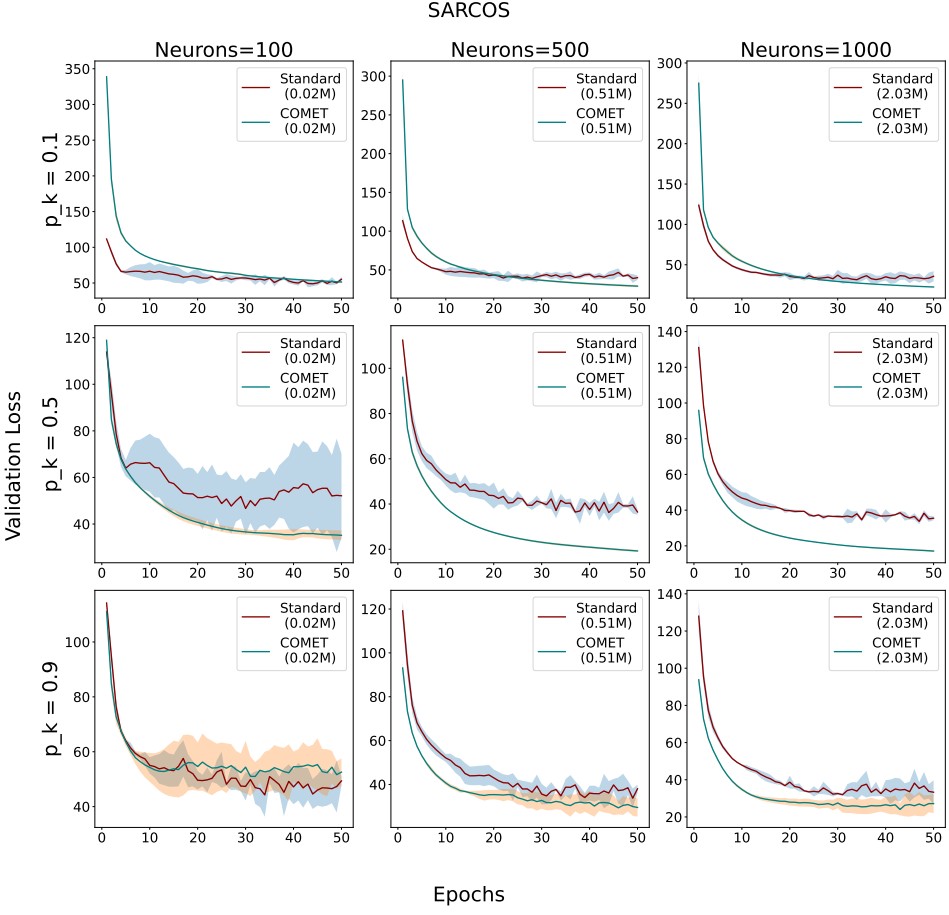

Figure 33: Illustration of 4-layers MLP trained on SARCOS.

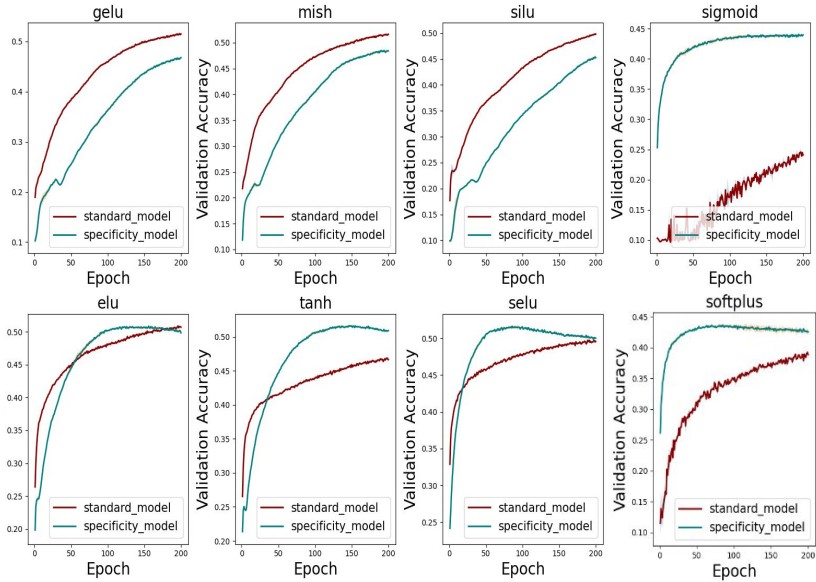

Figure 34: Comparison of the performance of COMET-trained MLP networks on the CIFAR10 dataset using different activation functions.

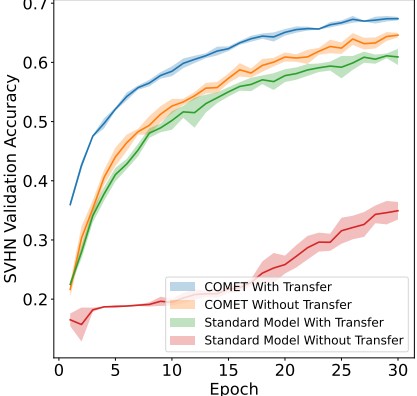

Figure 35: Non-IID Transfer: Evaluating COMET and the standard model on non-iid transfer learning. In particular, we train both COMET and the standard model on CIFAR10 followed by finetuning on SVHN (with transfer), or just train both models on SVHN (without transfer).

