# OpenReview forum: "More Experts Than Galaxies: Conditionally-Overlapping Experts with Biologically-Inspired Fixed Routing"
_ICLR.cc/2025/Conference — ICLR 2025 Poster_

### Official Review · Reviewer_rVeg · 2024-10-28

**Soundness:** 2
**Presentation:** 2
**Contribution:** 2
**Rating:** 3
**Confidence:** 4

**Summary:**

The paper, titled Conditionally Overlapping Mixture of Experts (COMET), proposes a method aimed at overcoming limitations in existing sparse neural network architectures. COMET introduces a modular, sparse structure with a biologically inspired fixed routing approach that eliminates the need for task IDs and trainable gating functions, commonly associated with representation collapse and redundancy. Instead, the authors implement a k-winner-take-all cap operation, enabling experts to overlap based on input similarity. This approach aims to improve generalization and facilitate faster learning, validated across various tasks and architectures, including MLPs, Vision Transformers, and GPT-based models.

**Strengths:**

1. COMET presents a novel routing method that replaces trainable gating functions with fixed, biologically inspired routing, which is rare in modular neural network approaches.

2. The proposed method is tested across diverse architectures and tasks, such as image classification, language modeling, and regression, suggesting versatility.

**Weaknesses:**

1. Section 3 lacks crucial methodological details necessary for a complete understanding of the proposed COMET approach. For instance, it is unclear which specific design elements in COMET were directly influenced by the concept of biological random projections.

2. The paper’s writing lacks clarity, making it difficult to fully understand the design of COMET. I recommend including a preliminary section that outlines the foundational Mixture of Experts (MoE) framework, followed by a clear discussion on how COMET’s design diverges from and improves upon existing methods.

3. While COMET is designed to improve modularity and interpretability, the authors do not demonstrate how the model’s interpretability has improved. More extensive interpretability metrics or qualitative evaluations would support the claimed benefits of COMET.

4. The absence of code and essential implementation details significantly hampers reproducibility and raises concerns about the robustness of the results.

**Questions:**

1. Which components of COMET are inspired by the concept of biological random projection?

2. How should the hyperparameter $k$ in Equation (3) be determined?

---

> ### Author Response · Authors · 2024-11-21
> **Rebuttal by Authors**
>
> We thank the reviewer for their feedback on our work. The reviewer raises a very interesting methodological/modeling question, and identifies a reference issue, both of which we address below.
>
> 1) Section 3 lacks crucial methodological details necessary for a complete understanding of the proposed COMET approach. For instance, it is unclear which specific design elements in COMET were directly influenced by the concept of biological random projections.
> * Thank you for pointing out the need for more methodological details in Section 3. Per your 2nd comment, we will add a preliminary section to review past work on MoE and input-dependent masking and explain how our concept of overlapping experts bridges these two literatures.
>
> * Regarding your Question 1 about the biological influence on COMET's design: the two biological mechanisms we draw inspiration from are random projections and k-winner-take-all capping. These mechanisms work together in our routing network to determine the mask at each layer for a given input. In the brain, a random projection plus capping leads to representations with low overlap between distinct inputs except when those inputs are similar (Bruhin & Davies, 2022). This property arises in our architecture for similar reasons (Fig 2). We will add this explanation to the revision. We would also like to clarify that these concepts simply serve as a motivation for our approach. We believe that drawing inspiration from biological systems can be a strength, rather than a weakness.
>
> * Nevertheless, we originally discussed the biological motivation in the Introduction and Related Work sections. Specifically, we mentioned in the Introduction, "we employ a k-winner-take-all cap operation, inspired by the brain’s efficient use of a limited number of active cells via lateral inhibition. This design choice is both biologically motivated and automatically determined through fixed random projections." Additionally, in the Related Work section, we noted that "we propose sparsification using a biologically motivated approach of random projection followed by a cap operation, which activates the strongest cells in the network, similar to the sensory system of fruit flies (Bruhin & Davies, 2022)."
>
> * To further clarify, we take biological inspiration for three ideas: 1) random projection; 2) sparsity; 3) k-winner-take-all. The biological inspiration from random projections is reflected in two specific design elements of COMET: the fixed random projections and the cap operation, which results in sparsity using the k-winner-take-all mechanism. We hope this clarification helps in addressing the reviewer’s concern and provides a clearer understanding of the role of biological inspiration in COMET's design.
>
> 2) The paper’s writing lacks clarity, making it difficult to fully understand the design of COMET. I recommend including a preliminary section that outlines the foundational Mixture of Experts (MoE) framework, followed by a clear discussion on how COMET’s design diverges from and improves upon existing methods.
> * We appreciate your feedback on the paper's clarity. We agree that a more comprehensive introduction to the concept of MoE followed by a clear discussion on how COMET differentiates from and improves upon existing methods would significantly enhance the paper. We will make sure to address this point more clearly in the revision.
>
> 3) While COMET is designed to improve modularity and interpretability, the authors do not demonstrate how the model’s interpretability has improved. More extensive interpretability metrics or qualitative evaluations would support the claimed benefits of COMET.
> * We acknowledge that our paper does not provide quantitative evidence of improved interpretability of the models, and that is because it was not our intention to demonstrate this aspect. The primary goal of COMET is to address specific challenges in existing work, such as: (1) trainable gating functions that lead to representation collapse, (2) non-overlapping experts that result in redundant computation and slow learning, and (3) reliance on explicit input or task IDs that limit flexibility and scalability. We understand that the references to interpretability in the paper may be misleading and we will remove them to avoid confusion.
>
> 4) The absence of code and essential implementation details significantly hampers reproducibility and raises concerns about the robustness of the results.
>
> * We understand the importance of reproducibility and transparency in research. In response to your comment, we have added our code as supplementary material for the majority of our experiments. We agree that this openness will not only facilitate the reproduction of our results but also provide a clearer understanding of our methodology, thereby addressing concerns about the robustness of our findings. We hope that this additional information will alleviate your concerns and allow for a more comprehensive evaluation of our work.

---

> ### Author Response · Authors · 2024-11-21
> **Rebuttal by Authors**
>
> 5) How should the hyperparameter “k” in Equation (3) be determined?
> * The value of the hyperparameter 'k' in Equation (3) can be determined experimentally, for instance, through its effect on cross-validated evaluation metrics. However, we found that performance is remarkably robust across a range of values of  'k'. For instance, as shown in Figure 4, COMET's performance remains consistent and outperforms the standard model when 'k' is set to 0.5 and 0.9 for neurons=1000, and 0.1, 0.5, and 0.9 for neurons=3000. This suggests that the exact value of 'k' is not crucial for achieving good performance. More importantly, our experiments (Sections 5.2.1, 5.2.2, 5.3, and 5.4) demonstrate that the performance gap between COMET-based models and their standard counterparts widens as the model size increases. This indicates that selective neuron activation becomes increasingly beneficial as network capacity grows. Therefore, rather than focusing on fine-tuning the value of 'k', it is more important to prioritize having a system with sufficient capacity to learn tasks effectively.

---

> > ### Comment · Reviewer_rVeg · 2024-11-26
> > **Thank You For the Rebuttal**
> >
> > Thank you for your reply and for providing the code in the supplementary. However, my concerns regarding the writing and interoperability have not been well addressed. I would like to keep my original score.

---

> > > ### Author Response · Authors · 2024-11-26
> > >
> > > We appreciate your feedback and are finishing up extensive revisions that should address your comments much more thoroughly. We hope you will reassess our paper when we post the revision. Thank you!

---

> > > > ### Author Response · Authors · 2024-11-27
> > > > **Summary of revisions**
> > > >
> > > > Thank you again for your constructive feedback. As you will see in our revised manuscript, we have thoroughly addressed the four weaknesses you identified:
> > > > * We explicitly state which of COMET’s design elements are biologically inspired (p. 2: 83-86) and we identify the elements of our formalism that correspond to these mechanisms (p. 4: 202-203).
> > > > * We have added a preliminary section 3 detailing standard MoE architectures and methods based on input-dependent masking. This section explains how COMET and the concept of overlapping MoE arises as a synthesis of these two lines of work and has been carefully integrated with the existing section 4. We also highlight the key differences between our approach and previous work (p. 5: 265-269).
> > > > * We have removed the references to interpretability in section 1, and we apologize for the impression that this was a goal of the research. We have also clarified the distinction between our present contributions and planned future work (p. 2: 98-102).
> > > > * We have added a placeholder for linking to our code repository (p. 2: 106) matching the supplement we recently uploaded.

---

> ### Author Response · Authors · 2024-12-01
> **Looking forward to your feedback on our response**
>
> Dear Reviewer rVeg,
>
> We sincerely appreciate your insightful and valuable comments. Given the limited time for the author-reviewer discussion phase, we are eagerly awaiting your further feedback. We hope the detailed explanations and the revised manuscript we have provided address the concerns in your review and affirm the merits of our paper. If you have any further inquiries or need additional clarification, please do not hesitate to reach out. We would be pleased to provide additional responses to further substantiate the efficacy of our research.
>
> Best Regards, Authors

---

### Official Review · Reviewer_ETwf · 2024-11-02

**Soundness:** 3
**Presentation:** 2
**Contribution:** 3
**Rating:** 8
**Confidence:** 4

**Summary:**

This paper introduces COMET (Conditionally Overlapping Mixture of ExperTs), a new method for creating sparse neural networks. The authors show using fixed, biologically-inspired routing can create more efficient and effective neural networks, particularly for larger models, while avoiding common problems in sparse architectures like representation collapse and poor knowledge transfer. The key insight is that COMET creates input-dependent sparsity without needing to learn the routing mechanism. COMET uses a fixed, biologically-inspired random projection combined with a k-winner-take-all operation to route inputs through the network, rather than using trainable gating functions.

**Strengths:**

Unlike other methods, the proposed COMET method has no trainable gating functions (unlike standard Mixture of Experts) and avoids representation collapse.

Does not require explicit input/task IDs or pre-defined expert specialization.

Works across multiple architectures (MLPs, ViTs, GPT, MLP-Mixers).

The work is particularly similar to \cite{cheung2019superposition}, especially with the use of a random projection matrix V to handle the decision to mask. The justifications for using random projections in \cite{cheung2019superposition} seem to align well with the described capacity benefits of the COMET method in larger networks as compared to smaller networks. In particular, with a larger number of neurons, the probability of interference between masks rapidly decreases.

@article{cheung2019superposition,
  title={Superposition of many models into one},
  author={Cheung, Brian and Terekhov, Alexander and Chen, Yubei and Agrawal, Pulkit and Olshausen, Bruno},
  journal={Advances in neural information processing systems},
  volume={32},
  year={2019}
}

**Weaknesses:**

The arguments for the number of experts is based on the possible permutations of masks that can be created which gives an unrealistically large number of possible experts. But this does not account for interference issues and establishing a bound more grounded in reality would be very helpful. The theory work in \cite{cheung2019superposition} should help better define these bounds.

There's a claim of "improved generalization through enhanced forward transfer", but it's unclear what experiments in this paper demonstrates better transfer learning.

**Questions:**

Is there any reason to believe this phenomenon does not already occur in large networks? \cite{elhage2022toy} describe a situation where neural networks encode the phenomenon observed in \cite{cheung2019superposition} during the course of training. Are there advantages of explicitly creating the superposition?

Figure 4 shows a strange result where the smaller_model performs consistently as well as the standard_model even for fairly low p_k values. It's unclear to me why there would be a benefit for COMET if at the neurons=3000, the smaller_network at pk=.1 will perform as well as the standard_model.  What is being gained here?

@article{elhage2022toy,
  title={Toy models of superposition},
  author={Elhage, Nelson and Hume, Tristan and Olsson, Catherine and Schiefer, Nicholas and Henighan, Tom and Kravec, Shauna and Hatfield-Dodds, Zac and Lasenby, Robert and Drain, Dawn and Chen, Carol and others},
  journal={arXiv preprint arXiv:2209.10652},
  year={2022}
}

---

> ### Author Response · Authors · 2024-11-21
> **Rebuttal by Authors**
>
> We thank the reviewer for their helpful questions, comments and feedback on our work. We believe that the proposed revisions and clarifications in line with the responses below will improve the strength of the paper.
>
> 1) The arguments for the number of experts is based on the possible permutations of masks that can be created which gives an unrealistically large number of possible experts. But this does not account for interference issues and establishing a bound more grounded in reality would be very helpful. The theory work in \cite{cheung2019superposition} should help better define these bounds.
> * We agree that this formulation is a bit tricky. The first reference you shared—\cite{cheung2019superposition}—specifically mentions that “a thorough analysis of how many different models can be stored in superposition with each other will be very useful.” This issue becomes particularly challenging when, for example, two experts with N neurons share (N−1) neurons. In this case, can we truly count them as distinct experts?
>
> * From a model-output perspective, the outputs of two such subsets of neurons could clearly differ. In this sense, we believe they should be considered different experts. From a more mathematical perspective, the second reference you shared—\cite{elhage2022toy}—discusses the Johnson–Lindenstrauss lemma, noting that “although it's only possible to have N orthogonal vectors in an N-dimensional space, it's possible to have exp⁡(N) many "almost orthogonal" vectors (based on cosine similarity) in high-dimensional spaces.”
>
> * That said, we believe this is somewhat tangential to the main focus of our paper, which is to demonstrate how such overlap is crucially beneficial for forward knowledge transfer, as shown in our experiments. We do recognize that other readers may have similar questions, and we will make sure to address this point more clearly in the paper.
>
> 2) There's a claim of "improved generalization through enhanced forward transfer", but it's unclear what experiments in this paper demonstrates better transfer learning.
> * We apologize for any confusion and would like to clarify. We recognize that transfer learning can be evaluated in different ways, with one common approach being pretraining on a large dataset followed by fine-tuning on smaller, domain-specific dataset. However, we focus on evaluating out-of-sample generalization—assessing how well models perform on a test set after being trained on a separate training set.
>
> * In Sections 4.1.1, 4.2, 4.3, 4.4, and many of the appendix experiments, we demonstrate how COMET improves out-of-sample generalization. We understand that this distinction may be unclear, and we will revise the paper to make this more explicit.
>
> 3) Is there any reason to believe this phenomenon does not already occur in large networks? \cite{elhage2022toy} describe a situation where neural networks encode the phenomenon observed in \cite{cheung2019superposition} during the course of training. Are there advantages of explicitly creating the superposition?
> * If we understand correctly, the two papers you referenced define “superposition” in different ways. In \cite{elhage2022toy}, superposition is described as “how and when models represent more features than they have dimensions,” whereas \cite{cheung2019superposition} frames it as "the ability to store multiple models within a single parameter instance." We would like to address your concern but, in order to do so, we would ask you to please clarify what you mean by “this phenomenon”.
>
> * If we may conjecture here, are you asking whether large networks inherently have input-dependent separation, and if so, what are the benefits of explicitly creating gates? If that’s the case, Section 4.1.1 shows that COMET’s unique sparsity improves kernel sharpness, leading to better generalization. Additionally, our experiments demonstrate that explicitly creating masks significantly enhances both performance and learning speed. We also found that COMET’s fixed masks outperform trainable ones in this context.

---

> > ### Comment · Reviewer_ETwf · 2024-11-25
> >
> > Appreciate the response. For the sake of interactive discussion, I'll refer to specific points.
> >
> > > If we understand correctly, the two papers you referenced define “superposition” in different ways. In \cite{elhage2022toy}, superposition is described as “how and when models represent more features than they have dimensions,” whereas \cite{cheung2019superposition} frames it as "the ability to store multiple models within a single parameter instance." We would like to address your concern but, in order to do so, we would ask you to please clarify what you mean by “this phenomenon”.
> >
> > For any representation generated from a linear layer, the superposition of features is the same as the superposition of weights. The phenomenon is studied as a natural property in LLMs \cite{elhage2022toy} that multiple unrelated features can coexist whereas \cite{cheung2019superposition} develop a procedure to explicitly combine unrelated features without interference.

---

> > > ### Author Response · Authors · 2024-11-27
> > > **Thanks for the further discussion**
> > >
> > > We agree superposition can be expressed equivalently in terms of weights or features, and indeed Cheung et al. present it both ways. We would like to point out (a) Cheung et al. superpose models for different tasks while Elhage et al. superpose features, and (b) Cheung et al.'s method applies when each task has fewer features than the dimension of the activation space (so that different tasks can be rotated to orthogonal subspaces) while Elhage's superposition applies in the opposite situation where a task has more features than the dimension of the activation space (so that the features must be represented non-orthogonally). Still we take the point that Elhage's feature superposition can describe a situation where Cheung's model superposition is applied with more tasks than can fit orthogonally into the space.
> > >
> > > It is a great question whether networks spontaneously form structures like what COMET enforces, in which dissimilar inputs are processed by less-overlapping sets of neurons. For present purposes the critical observation is that COMET outperforms standard networks, so clearly there is an advantage to creating the superposition explicitly. We will add these remarks to the conclusions section.

---

> > > > ### Author Response · Authors · 2024-11-27
> > > > **Summary of revisions**
> > > >
> > > > Thanks again for your feedback. As you can see in our revised paper, we have added significant material to address all of your main points:
> > > > * Careful consideration of prior work on superposition, interference, and their implications for our overlapping expert framework (p. 3: 157-161; p. 4: 246-251; p. 10: 533-538)
> > > > * Better delineating our present contributions from planned future work (p. 2: 98-102) and adding a preliminary experiment demonstrating COMET's potential for transfer learning on sequential tasks (Appendix A.9)
> > > > * Explaining the significance of the finding in Figure 4 that COMET beats the Standard Model even when the Standard Model is no better than the Smaller Model

---

> ### Author Response · Authors · 2024-11-21
> **Rebuttal by Authors**
>
> 4) Figure 4 shows a strange result where the smaller_model performs consistently as well as the standard_model even for fairly low p_k values. It's unclear to me why there would be a benefit for COMET if at the neurons=3000, the smaller_network at pk=.1 will perform as well as the standard_model. What is being gained here?
> * We would also appreciate further clarification here, if we may. In the subfigure you mentioned (neurons=3000, pk=0.1), COMET (in red) significantly outperforms both the smaller_network (in orange) and the standard_model (in blue). As you correctly observed, the smaller_network performs similarly to the standard_model, which is exactly the point we aimed to highlight in the figure. Specifically, reducing the number of neurons (to pk*N in the smaller_network, which matches the number of active neurons in COMET) does not lead to better performance—it performs just as well as the standard_model (excessively overparameterized network). Moreover, simply increasing the number of neurons, as in the standard_model, does not improve performance either. The key insight here is that COMET’s unique gating mechanism facilitates positive knowledge transfer, leading to faster learning and improved generalization.

---

> ### Author Response · Authors · 2024-12-01
> **Looking forward to your feedback on our response**
>
> Dear Reviewer ETwf,
>
> We sincerely appreciate your insightful and valuable comments. Given the limited time for the author-reviewer discussion phase, we are eagerly awaiting your further feedback. We hope the detailed explanations and the revised manuscript we have provided address the concerns in your review and affirm the merits of our paper. If you have any further inquiries or need additional clarification, please do not hesitate to reach out. We would be pleased to provide additional responses to further substantiate the efficacy of our research.
>
> Best Regards, Authors

---

> > ### Comment · Reviewer_ETwf · 2024-12-02
> >
> > Overall, I'm happy with the updates and this is an interesting method of creating sparsity without the need to train a router. Therefore, I have raise my score.

---

### Official Review · Reviewer_fx4N · 2024-11-03

**Soundness:** 3
**Presentation:** 3
**Contribution:** 2
**Rating:** 6
**Confidence:** 3

**Summary:**

The paper introduces Conditionally Overlapping Mixture of ExperTs (COMET).
COMET uses biologically inspired, fixed random projections to generate binary masks that define subnetworks know as 'experts'.
The mask generation process is input-dependent, causing similar inputs to activate overlapping sets of experts.
The authors test the models on a range of benchmark tasks, finding that COMET performs well, particularly for large model sizes.

**Strengths:**

The paper is well written and the core idea is explained clearly.

The authors demonstrate key properties of the COMET model, notably showing that similar inputs tend to activate overlapping experts, facilitated by the fixed gating mechanism.

The model is tested on a wide selection of benchmark tasks including computer vision, language modelling, and regression.

The authors demonstrate the benefit of using COMET, particularly at large model sizes.

The use of COMET requires no additional trainable parameters which is quite advantageous.

**Weaknesses:**

In other works these gating functions can help alleviate catastrophic forgetting for tasks that are presented sequentially, but this is something that has not been tested in this paper.

Given that previous work has similarly employed networks to determine gating, I am not entirely convinced that the novelty here is sufficient. However, I acknowledge that unlike prior approaches, which relied on trainable gates, this method uses fixed random projections.

There will be additional computational costs to using COMET but there is not an in-depth analysis of these costs in the paper. An analysis of training/inference times and GPU memory usage between COMET and the standard models would strengthen the submission.

**Questions:**

How does this model perform on a continual learning benchmark, such as permuted MNIST or split-CIFAR-100?

What are the additional costs to using COMET in terms of training time and memory usage?

---

> ### Author Response · Authors · 2024-11-21
> **Rebuttal by Authors**
>
> We thank the reviewer for their feedback and questions. We believe that the concerns raised can be addressed directly in this reply, or through the revisions of the paper, and address them below:
>
> 1) In other works these gating functions can help alleviate catastrophic forgetting for tasks that are presented sequentially, but this is something that has not been tested in this paper.
>
> * We completely agree that addressing catastrophic forgetting is a crucial area of research, and we acknowledge the work done in this space. In fact, the potential applicability to continual learning is one of the primary motivations behind our work. Your observation that other gated sparse methods have succeeded in CL suggests that COMET may have promise there as well. Moreover, there is reason to believe COMET's unique features will lead to gains over existing CL approaches. Beyond the reduced variance from COMET's fixed routing function, we have highlighted how similar inputs tend to share more parameters, which facilitates positive knowledge transfer and improved generalization. Note that this approach of encouraging overlap is radically different from popular methods in continual learning, which try to reduce the overlap [4]. However, extending the evaluation to CL settings would require a very substantial addition to the paper, which we believe would be beyond its scope. The aim of this initial paper is to present the basic method and establish its advantage on single tasks. This strategy parallels previous work: none of the original mixture of expert papers [1, 2] were originally evaluated as continual learning algorithms, nor was the popular sparse MoE paper [3], which also uses gating functions. In our revision we will more clearly delineate the contributions of the present paper from planned future work.
>
> **References:**
>
> 1) @ARTICLE{Adaptive, author={Jacobs, Robert A. and Jordan, Michael I. and Nowlan, Steven J. and Hinton, Geoffrey E.}, journal={Neural Computation}, title={Adaptive Mixtures of Local Experts}, year={1991}, volume={3}, number={1}, pages={79-87}, keywords={}, doi={10.1162/neco.1991.3.1.79}}
> 2) @INPROCEEDINGS{Hierarchical, author={Jordan, M.I. and Jacobs, R.A.}, booktitle={Proceedings of 1993 International Conference on Neural Networks (IJCNN-93-Nagoya, Japan)}, title={Hierarchical mixtures of experts and the EM algorithm}, year={1993}, volume={2}, number={}, pages={1339-1344 vol.2}, keywords={Machine learning algorithms;Surface fitting;Vectors;Supervised learning;Mars;Orbital robotics;Biological neural networks;Jacobian matrices;Psychology;Partitioning algorithms}, doi={10.1109/IJCNN.1993.716791}}
> 3) @inproceedings{ shazeer2017, title={ Outrageously Large Neural Networks: The Sparsely-Gated Mixture-of-Experts Layer}, author={Noam Shazeer and *Azalia Mirhoseini and *Krzysztof Maziarz and Andy Davis and Quoc Le and Geoffrey Hinton and Jeff Dean}, booktitle={International Conference on Learning Representations}, year={2017}, url={https://openreview.net/forum?id=B1ckMDqlg} }
> 4) @misc{cheung2019superpositionmodels, title={Superposition of many models into one}, author={Brian Cheung and Alex Terekhov and Yubei Chen and Pulkit Agrawal and Bruno Olshausen}, year={2019}, eprint={1902.05522}, archivePrefix={arXiv}, primaryClass={cs.LG}, url={https://arxiv.org/abs/1902.05522}, }

---

> ### Author Response · Authors · 2024-11-21
> **Rebuttal by Authors**
>
> 2) Given that previous work has similarly employed networks to determine gating, I am not entirely convinced that the novelty here is sufficient. However, I acknowledge that unlike prior approaches, which relied on trainable gates, this method uses fixed random projections.
>
> * Most of the ingredients of COMET appear in prior methods. We believe our main contribution is the integration of concepts from diverse research areas into a concise framework that addresses the research problem. COMET combines the ideas of fixed random projection and k-winner-take-all from neuroscience, routing functions from modular neural networks, expert-based approaches from the MoE literature, the notion of implicit experts from dynamic neural networks, the integration of sparsity and modularity from conditional computation, input-dependent masking from various deep learning areas, and the importance of active parameter overlap from continual learning, which led us to coin the term “overlapping experts”. By bringing these concepts together, COMET addresses many challenges that existing work has:
>
> * First, COMET solves the issue of trainable gating functions, which are notoriously difficult to train and often lead to representation collapse. Second, unlike many prior approaches, COMET leverages overlapping experts. Third, COMET ensures that similar inputs are consistently mapped to the same experts—a problem that has not been effectively solved in previous work—which we hypothesize will facilitate forward knowledge transfer, even without supervision. Additionally, COMET does not rely on input or task IDs to determine which gating mask to apply, offering greater flexibility and scalability. Fourth, COMET supports an exponentially large number of (overlapping) experts, overcoming the limitations of methods that work with only a small number of experts, which may be insufficient for more complex tasks.
>
> * Through a comprehensive ablation study, we demonstrate that addressing all of these challenges simultaneously is crucial for achieving the benefits of COMET. Specifically, we show that removing or modifying individual components of COMET, such as its routing mechanism, sparsity, or modularity, leads to significant degradation in performance. For example, we find that simply increasing or reducing the number of parameters or experts, or replacing the fixed gating function by a trainable one, is not sufficient to achieve the same level of improvement. Our results suggest that the synergistic combination of these elements in COMET is essential for achieving improved learning speed and generalization across a wide range of benchmarks and architectures, while using fewer active parameters.
>
> 3) There will be additional computational costs to using COMET but there is not an in-depth analysis of these costs in the paper. An analysis of training/inference times and GPU memory usage between COMET and the standard models would strengthen the submission.
>
> * We appreciate the suggestion to analyze computational costs and in our revision we will add such analysis.

---

> > ### Comment · Reviewer_fx4N · 2024-11-26
> >
> > I thank the authors for their work during the discussion phase. I appreciate that a great deal of time has clearly gone into producing and strengthening the paper.
> >
> > After the discussion phase, I still hold my initial views about the weaknesses of the paper, which are difficult to address in a short timeframe. I will keep my score at a 6.

---

> > > ### Author Response · Authors · 2024-11-27
> > > **Summary of revisions**
> > >
> > > Thanks again for your feedback. As you can see in our revised paper, we have added significant material to address all three of your main points:
> > > * We added an experiment evaluating COMET’s running time and memory usage (Appendix A.8).
> > > * We explain the critical differences from previous similar approaches (p. 2: 93-98; p. 5: 265-269).
> > > * We better delineate our present contributions from planned future work, including catastrophic forgetting (p. 2: 98-102). We have also added a preliminary experiment demonstrating COMET's potential for transfer learning on sequential tasks (Appendix A.9). Although this experiment does not assess forgetting, the results show the promise of COMET's approach for sequential data distributions, which is one of the key metrics for models’ performance on continual learning settings.

---

> ### Author Response · Authors · 2024-12-01
> **Looking forward to your feedback on our response**
>
> Dear Reviewer fx4N,
>
> We sincerely appreciate your insightful and valuable comments. Given the limited time for the author-reviewer discussion phase, we are eagerly awaiting your further feedback. We hope the detailed explanations and the revised manuscript we have provided address the concerns in your review and affirm the merits of our paper. If you have any further inquiries or need additional clarification, please do not hesitate to reach out. We would be pleased to provide additional responses to further substantiate the efficacy of our research.
>
> Best Regards, Authors

---

> > ### Comment · Reviewer_fx4N · 2024-12-02
> >
> > Dear Authors,
> >
> > I appreciate the great effort that has gone into the paper during this discussion phase. While I believe the submission has been strengthened and some weaknesses have been addressed, some issues remain. As such, I do not believe the paper's score should be raised to an 8, and I will retain my score of 6.
> >
> > My primary criticism is that the contribution's strength over prior work is not sufficient to warrant an 8, which is why I am hesitant to raise the score. Although the authors have argued eloquently for the work's novelty, I continue to hold this view.
> >
> > I hope the authors do not interpret my decision to maintain the score of 6 as a negative reaction to the additional work provided, which has been both interesting and insightful to review.
> >
> > Once again, I thank the authors for their hard work.

---

### Author Response · Authors · 2024-11-21
**Author Rebuttal**

We would like to thank the reviewers for their thoughtful feedback, and were pleased to see agreement on the following positive points:

**Importance & Novelty:**

COMET is a novel method for creating sparse neural networks. It achieves input-dependent sparsity using a fixed, biologically inspired routing mechanism in place of the trainable gates used by prior methods. In more detail:

* fx4N: “unlike prior approaches, which relied on trainable gates, this method uses fixed random projections.”
* ETwf: “a new method for creating sparse neural networks (...) “COMET creates input-dependent sparsity without needing to learn the routing mechanism (...) Unlike other methods, the proposed COMET method has no trainable gating functions”
* rVeg: “COMET presents a novel routing method that replaces trainable gating functions with fixed, biologically inspired routing, which is rare in modular neural network approaches”.

**Key Properties of COMET:**

Similar inputs tend to activate overlapping experts, without the need for task IDs. In more detail:

* fx4N: “similar inputs tend to activate overlapping experts, facilitated by the fixed gating mechanism”.
* ETwf: “Does not require explicit input/task IDs or pre-defined expert specialization”
* rVeg:  “biologically inspired fixed routing approach that eliminates the need for task IDs and trainable gating functions

**Efficiency and Effectiveness:**

COMET shows improved performance relative to baseline approaches, especially for larger models, while avoiding representation collapse. In more detail:

* fx4N: “COMET requires no additional trainable parameters which is quite advantageous. (...) “The authors demonstrate the benefit of using COMET, particularly at large model sizes”
* ETwf: “The authors show using fixed, biologically-inspired routing can create more efficient and effective neural networks, particularly for larger models, while avoiding (...) representation collapse and poor knowledge transfer”
* rVeg: “eliminates the need for task IDs and trainable gating functions, commonly associated with representation collapse and redundancy”

**Versatility and generalization:**

COMET works well across diverse architectures and tasks. In more detail:

* fx4N: “The model is tested on a wide selection of benchmark tasks including computer vision, language modelling, and regression”.
* ETwf: “Works across multiple architectures (MLPs, ViTs, GPT, MLP-Mixers)”.
* rVeg: “The proposed method is tested across diverse architectures and tasks, such as image classification, language modeling, and regression”.

There were also concerns raised by more than one reviewer in two different areas: experimental evaluation and distinction from existing methods. We group these below, but given that each concern touches on a different point, we address these through individual reviewer responses.

**Experimental Evaluation:**

* fx4N: “these gating functions can help alleviate catastrophic forgetting for tasks that are presented sequentially, but this is something that has not been tested in this paper”.
* ETwf: “There's a claim of "improved generalization through enhanced forward transfer", but it's unclear what experiments in this paper demonstrates better transfer learning”.
* rVeg:  “While COMET is designed to improve modularity and interpretability, the authors do not demonstrate how the model’s interpretability has improved”.

**Distinction from Existing Methods:**

* fx4N: “Given that previous work has similarly employed networks to determine gating, I am not entirely convinced that the novelty here is sufficient”
* rVeg: “I recommend including a preliminary section that outlines the foundational Mixture of Experts (MoE) framework, followed by a clear discussion on how COMET’s design diverges from and improves upon existing methods.”

---

### Author Response · Authors · 2024-11-27
**Revision Posted**

We have just posted our revised paper. All new text is temporarily marked in orange. Please see the following major additions which respond to all the main points in the reviews. We hope you’ll agree the paper is now significantly strengthened.
* Role of biological inspiration (p. 2: 83-86)
    * Random projections, $k$-winner-take-all capping, and sparse representations with overlap that depends on input similarity
    * Correspondence between these mechanisms and elements of our formalism (p. 4: 202-203)
* Novelty and relation to previous work
    * Key ideas COMET brings together from several research areas (p. 2: 93-98)
    * Critical differences from other methods (p. 5: 265-269)
* Delineate present contributions from future work (p. 2: 98-102)
    * We clarify that transfer learning, continual learning, and catastrophic forgetting are promising directions but not studied here (except see sec A.9)
    * We have removed mention of interpretability which was never a goal
* Link to code repo (placeholder) (p. 2: 106)
* Relationship to model superposition of Cheung et al. (2019) (p. 3: 157-161)
* Preliminary section detailing MoE and input-dependent masking (p. 4: sec 3)
    * COMET and our proposal of overlapping experts synthesize these two literatures
* Considerations for number of experts (p. 4: 246-251)
    * Exponentially many experts are possible even with bounded interference (Elhage et al., 2022)
    * More importantly, ‘interference’ is desirable with our similarity-dependent routing
* Explained the significance of COMET beating the Standard Model when the Standard Model cannot beat the Smaller Model (p. 8: 424-426)
* Possibility that superposition emerges naturally in large networks (p. 10: 533-538)
    * If so, COMET still adds an advantage by explicitly imposing this structure
* Analysis of running times and memory usage (pp. 23-25, sec A.8)
* Preliminary test of transfer learning (pp. 26-27, sec A.9)

---

### Meta-Review · Area_Chair_r3Di · 2024-12-21

**Metareview:**

**Summary**

The paper introduces Conditionally Overlapping Mixture of Experts (COMET), a novel method for enhancing sparse neural networks by using biologically inspired, fixed random projections to generate input-dependent binary masks. These masks define subnetworks called 'experts' whose activation overlaps based on input similarity, using a k-winner-take-all mechanism for routing inputs. This approach avoids the need for task IDs and trainable gating functions, which are common sources of issues like representation collapse and redundancy in sparse architectures. COMET is shown to perform well, especially in larger models, and improves efficiency, generalization, and speed of learning across various benchmarks and neural network architectures, including MLPs, Vision Transformers, and GPT-based models.



**Strengths**


* COMET introduces a novel routing method that replaces trainable gating functions with fixed, biologically inspired routing, a unique approach in modular neural network designs.
* The proposed method has been tested on diverse architectures and tasks, including image classification, language modeling, and regression, demonstrating its merits.
* COMET does not require explicit input/task IDs or pre-defined expert specialization, moreover, it does not require additional trainable parameters.

**Weaknesses**

* In light of the prior work in the literature, the novelty of this paper is marginal.
* The argument for the number of experts, based on the potential permutations of masks creating an unrealistically large number, fails to consider interference issues.
* The paper lacks an analysis of the computational costs of the proposed algorithm.
* The paper suffers from overclaiming. For example, while the authors assert that COMET enhances modularity and interpretability, they fail to demonstrate any actual improvement in the model's interpretability.
* The lack of experiments in continual learning seems like a missed opportunity.

**Conclusion**

The paper elicited polarized reviews. Reviewers fx4N and ETwf viewed it positively, though they noted its limited novelty. Reviewer rVeg assigned a low rating of 3 with a confidence of 4, but their review did not substantiate this negative assessment. The authors responded with a substantial rebuttal and engaged positively with the reviewers. After thorough consideration of the paper, the reviews, and the rebuttal, I find the work to be marginally above the acceptance threshold and therefore vote to accept the paper.

**Additional Comments On Reviewer Discussion:**

During the discussion period, reviewers fx4N and ETwf engaged positively, focusing on the paper's novelty and highlighting missed opportunities such as continual learning. After reviewing the paper, the discussions, the reviews, and the authors' rebuttal, I find that the strengths of the paper outweigh the weaknesses, leading me to vote in favor of accepting the paper.

---

### Decision · Program_Chairs · 2025-01-22

Accept (Poster)